# TIMEOMNI-VL: Unified Models for Time Series Understanding and Generation

Tong Guan[† * 1 2]   Sheng Pan[* 1]   Johan Barthelemy[3]   Zhao Li[4]   Yujun Cai[5]   Cesare Alippi[6 7]
Ming Jin[1]   Shirui Pan[1]

## Abstract

Recent time series modeling faces a sharp divide between numerical generation and semantic understanding, with research showing that generation models often rely on superficial pattern matching, while understanding-oriented models struggle with high-fidelity numerical output. Although unified multimodal models (UMMs) have bridged this gap in vision, their potential for time series remains untapped. We propose TIMEOMNI-VL, the first vision-centric framework that unifies time series understanding and generation through two key innovations: (1) Fidelity-preserving bidirectional mapping between time series and images (Bi-TSI), which advances Time Series-to-Image (TS2I) and Image-to-Time Series (I2TS) conversions to ensure near-lossless transformations. (2) Understanding-guided generation. We introduce TSUMM-SUITE, a novel dataset consisting of six understanding tasks rooted in time series analytics and coupled with two generation tasks. With a calibrated Chain-of-Thought (CoT), TIMEOMNI-VL is the first to leverage time series understanding as an explicit control signal for high-fidelity generation. Experiments confirm that this unified approach significantly improves semantic understanding and numerical precision, establishing a new frontier for multimodal time series modeling. Code[1] and checkpoint[2] are publicly available.

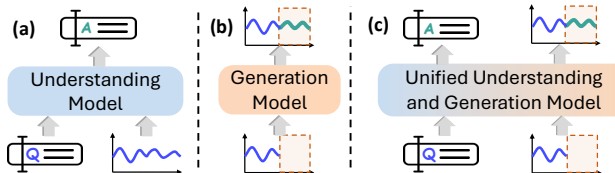

*Figure 1.* Comparison of architectures for **(a)** time series understanding model that produces textual answers only, **(b)** time series generation model that outputs time series only, and **(c)** unified time series understanding and generation model that supports both answering queries and generating time series.

## 1. Introduction

Time series are pervasive in modern systems and everyday life, underpinning decision-making across healthcare, transportation, industrial monitoring, and finance (Huang et al., 2026; Zou et al., 2025; wang et al., 2025; Ye et al., 2024). With advances in time series modeling at scale, recent progress has largely followed two parallel threads: (1) *Generation models.* Led by time series foundation models (TSFMs), this thread prioritizes high-fidelity numerical sequence generation, excelling in tasks such as forecasting (Shi et al., 2025) and data imputation (Goswami et al., 2024) (Figure 1b). (2) *Understanding Models.* Influenced by the rise of large language models (LLMs), this thread focuses on temporal reasoning (Guan et al., 2026) by providing explicit, human-readable interpretations of complex dynamics (Xie et al., 2025) (Figure 1a). However, a significant divide remains: Generation models often lack explicit structural understanding despite offering representation analysis on signal components (Wang et al., 2025b), while understanding-oriented models frequently struggle with high-fidelity numerical generation as text-native tokenizers can disrupt numerical continuity (e.g., "123" → "1", "2", "3"). Bridging this gap with a unified model capable of both understanding and generation represents an urgent need for time series processing (Figure 1c).

Likewise, the vision domain has undergone a similar trajectory, with models specialized for visual generation (Nichol et al., 2022; Razavi et al., 2019) and those focusing on visual understanding (Radford et al., 2021; Wang et al., 2024). However, recently, the vision community has witnessed advancements in unified multimodal models (UMMs) that

---

[†]Project lead [*]Equal contribution [1]Griffith University [2]Zhejiang University [3]NVIDIA [4]Zhejiang Lab [5]The University of Queensland [6]Università della Svizzera Italiana [7]Politecnico di Milano. Correspondence to: Ming Jin <mingjinedu@gmail.com>, Shirui Pan <s.pan@griffith.edu.au>.

*Proceedings of the $43^{rd}$ International Conference on Machine Learning*, Seoul, South Korea. PMLR 306, 2026. Copyright 2026 by the author(s).

[1] https://github.com/AntonGuan/TimeOmni-VL
[2] https://huggingface.co/TimeOmni-VL/TimeOmni-VL

excel in both image understanding and generation. A key emerging insight is that robust understanding serves as a foundation for superior generation, since structured semantic guidance improves controllability and fidelity (Zhang et al., 2025). Meanwhile, an emerging line of work suggests a similarity between time series and vision modality, as pixel-level variations in natural images can be viewed as sequential signals and exhibit intrinsic commonalities with time series (Chen et al., 2025b). By reframing time series as a visual inpainting problem, visual generative models (He et al., 2022) can achieve impressive time series forecasting (Shen et al., 2025) and imputation (Maaroufi et al., 2021; Noufel et al., 2025) performance even in a training-free manner. Despite their effectiveness, these vision-based approaches largely rely on superficial texture imitation rather than genuine temporal understanding. They lack a mechanism to interpret the underlying signal dynamics from a time series perspective, which includes identifying trend shifts or seasonal dependencies within the visual space. Motivated by these observations, we ask a natural question: *Is it possible to represent time series in the vision modality and thereby internalize time series understanding and generation as native capabilities of UMMs, so that time series performance improves naturally as UMMs continue to advance?*

However, achieving this integration is non-trivial as two fundamental challenges remain: **(1) Fidelity-preserving bidirectional mappings between time series and images are still lacking.** Although VisionTS-style (Chen et al., 2025b; Shen et al., 2025) converters offer a practical interface for vision models, we find that the front-end conversion itself can discard numerical information, so the model may not observe the complete series content. Once information is lost at the input stage, it cannot be recovered downstream, making high-fidelity generation fundamentally unattainable. **(2) Understanding-guided generation remains underexplored for time series.** While UMMs possess strong semantic capabilities, they are not yet grounded in time series properties such as inherent periodicity and structural change-points. As a result, they cannot leverage semantics to guide time series generation, preventing the system from achieving the precise and controllable results commonly observed in other multimodal tasks.

To address these challenges, we build TIMEOMNI-VL around two core design objectives: (1) Fidelity-preserving bidirectional mappings between time series and images and (2) understanding-guided generation (as our primary goal is precise generation, where understanding serves as the necessary control signal, not vice versa). We advance existing converters (Chen et al., 2025b) into a fidelity-oriented **Bi**directional **T**ime **S**eries ⇔ **I**mage mappings (**Bi-TSI**) that avoid information loss at the input stage. Concretely, we introduce robust fidelity normalization (RFN) to stabilize dynamic-range projection and preserve peak geometry un-

der realistic signals, alongside encoding capacity control to prevent implicit downsampling when rendering time series onto a fixed time series image (TS-image) canvas. Building on Bi-TSI, we construct a new dataset **TSUMM-SUITE** by specifying forecasting and imputation as generation tasks and deriving six understanding tasks from the same generation instances, organized into layout-level and signal-level analysis. These tasks encourage UMMs to interpret TS-images from a temporal perspective rather than relying on superficial textures. Finally, we present **TIMEOMNI-VL**, the first vision-centric framework that internalizes time series understanding and generation as native capabilities of UMMs. To enable understanding-guided generation, we form a generation Chain-of-Thought (CoT) by organizing the understanding QAs of each generation instance into a calibrated reasoning chain, making temporal understanding an explicit control signal for precise and controllable time series generation. Our contributions lie in three aspects:

**1. New Models.** We present **TIMEOMNI-VL**, the first vision-centric framework that unifies time series understanding and generation. TIMEOMNI-VL integrates: (1) Fidelity-preserving bidirectional Time Series ⇔ Image mappings to prevent implicit information loss. (2) Generation CoT that organizes instance-level understanding into a calibrated reasoning chain and serves as an explicit control signal for numerical generation tasks like forecasting and imputation.

**2. New Datasets and Testbed.** We introduce **TSUMM-SUITE**, a benchmark comprising two generation tasks and six understanding tasks. The understanding tasks are tailored to the TS-image representation produced by TIMEOMNI-VL, and are organized into **layout-level** and **signal-level** analyses to encourage temporal interpretation rather than superficial texture.

**3. Comprehensive Evaluation.** Results demonstrate that the understanding tasks effectively teach the base model to interpret TS-images: TIMEOMNI-VL boosts the base model from near-zero accuracy to near-perfect scores on four understanding tasks (approaching 1.0). On generation, TIMEOMNI-VL achieves top-tier results on forecasting and reaches state-of-the-art performance on imputation. Moreover, the proposed generation CoT consistently improves generation quality, yielding an average 8.2% gain.

## 2. Related Work

**Time Series Generation Models.** In this context, time series generation specifically refers to forecasting and imputation tasks rather than synthetic data generation. Existing models are primarily categorized into two paradigms. **(1) Time series-based models.** Early efforts focused on developing domain-specific architectures which often lacked cross-dataset generalization (Wu et al., 2020; Guan et al.,

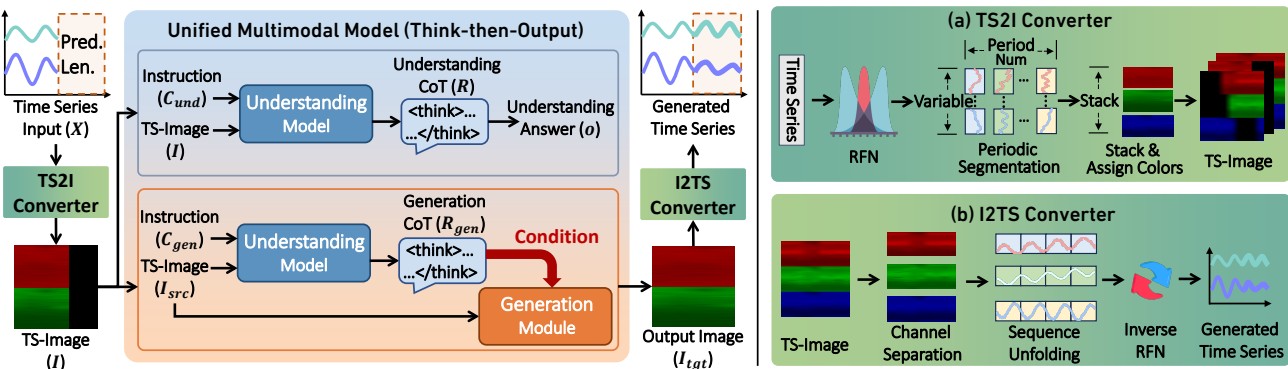

*Figure 2.* Overview of the TIMEOMNI-VL framework. The input time series is first converted into a TS-image $I$ by the **(a) TS2I Converter**. For understanding tasks, the understanding model directly produces CoT $R$ and the final answer. For generation tasks, the understanding model first generates CoT $R_{\text{gen}}$ as a condition for the generation module to generate the target image $I_{\text{tgt}}$, which is then converted back to a time series by the **(b) I2TS Converter**. Detailed pipelines of the TS2I and I2TS converters are shown on the right.

2025; Wang et al., 2025b). With the increasing availability of large-scale datasets, training TSFMs from scratch has become the mainstream approach to achieve superior zero-shot generalization (Woo et al., 2024; Ansari et al., 2024; Shi et al., 2025). **(2) Image-based models.** Early researchers explored convolutional (Wang & Oates, 2015) and patch-based (Maaroufi et al., 2021; Noufel et al., 2025) methods to reconstruct time series as images, revealing shared properties between the two modalities. Following the success of general visual generative models, the TS2I paradigm has resurged through models like VisionTS (Chen et al., 2025b; Shen et al., 2025), which demonstrate impressive zero-shot capabilities. However, their reliance on pixel-level pattern matching lacks genuine temporal understanding.

**Time Series Understanding Models.** Time-LLM (Jin et al., 2024) leverages the generalization capabilities of LLMs for time series, yet its understanding of temporal patterns remains largely implicit. To achieve explicit understanding, existing research has branched into two primary directions. The first involves **time series language models (TSLMs)**, which utilize synthetic datasets to align temporal signals with textual descriptions to ground temporal semantics (Xie et al., 2025; Kong et al., 2025; wang et al., 2025; Langer et al., 2025). The second encompasses **time series reasoning models (TSRMs)**, which leverage the R1-paradigm (DeepSeek-AI, 2025) to enhance temporal reasoning (Guan et al., 2026; Ni et al., 2026). Despite these advancements, both categories are constrained by the text-centric nature of LLMs. Standard vocabularies typically fragment multi-digit numbers into discrete tokens, thereby disrupting numerical continuity and undermining the precision required for high-fidelity generation.

**Unified Multimodal Models.** UMMs have recently emerged in the vision community to integrate understanding and generation within a single framework. These models generally follow either a unified auto-regressive architec-

ture (Team, 2025; Tong et al., 2025; Wu et al., 2025a; Chen et al., 2025a; Cui et al., 2025) or a hybrid paradigm combining auto-regression with diffusion (Ma et al., 2025; Deng et al., 2025; Wu et al., 2025b). Currently, the hybrid approach often yields superior results because image understanding prioritizes high-level semantics while generation requires fine-grained pixel details (Zhang et al., 2025; Deng et al., 2025). Since the time series community lacks universal pre-trained encoders equivalent to ViT (Dehghani et al., 2023) or VAE (Kingma & Welling, 2022) in vision, recent studies (Parker et al., 2025; Wu et al., 2026) attempting unified modeling with auto-regressive LLMs typically rely on shallow MLP layers. However, the effectiveness of such simple layers in projecting time series into the latent space remains unverified. This gap motivates combining TS2I methods with UMMs: by utilizing images as a modality-specific enhancement, we leverage UMMs to achieve a unified framework for temporal understanding and generation.

## 3. Methodology

In this section, we first establish a unified problem formulation for both tasks. We then present TIMEOMNI-VL, the first vision-centric framework that unifies time series understanding and generation. Finally, we introduce TSUMM-SUITE and its construction pipeline, which formalizes both generation and understanding tasks and bridges them by deriving generation CoT directly from understanding QAs.

**Problem Definition.** We formulate unified time series understanding and generation as a conditional *think-then-output* process within UMMs. Unlike in TSRMs (Guan et al., 2026), where CoT mainly serves as a textual explanation, here we treat CoT as a control signal that conditions generation. Given (1) the observed time series input $\mathbf{X} \in \mathbb{R}^{T \times N}$, and (2) an auxiliary context $C$ (e.g., task instructions), the model first generates a CoT $R = (r_1, \ldots, r_K)$,

*Figure 3.* Illustration of improvements in Bi-TSI. (a) **Robust fidelity normalization** enables near-lossless rendering of **high-dynamic-range time series** by keeping values within the valid pixel range, whereas the baseline in VisionTS++ (Shen et al., 2025) can overflow this range and fail to represent the spike. (b) **Encoding capacity control** prevents implicit downsampling when encoding **high-dimensional time series**, ensuring that the resulting TS-image remains information-preserving, whereas the baseline suffers information loss.

and then produces the task target $o$ using $R$ as additional context. Formally,

$$p_\theta(R, o \mid \mathbf{X}, C) = p_\theta(R \mid \mathbf{X}, C)p_\theta(o \mid R, \mathbf{X}, C). \quad (1)$$

To standardize the inference process, we explicitly instruct the model to enclose the CoT $R$ within `<think></think>` tags across all tasks.

In this paper, we transform time series into the TS-image $I = \mathcal{V}(\mathbf{X})$. For understanding tasks on the TS-image $I$, the output produces a textual answer. For generation tasks (e.g., forecasting or imputation), we formulate the problem as editing the input TS-image: given a source image $I_{\text{src}}$ and a generation instruction $C_{\text{gen}}$, the model outputs an edited image $I_{\text{tgt}}$, which is then decoded back into numerical values.

**Overall Framework.** As illustrated in Figure 2, we design TIMEOMNI-VL to handle both time series understanding and generation tasks. We use Bagel (Deng et al., 2025) as the backbone UMM. While our framework is backbone-agnostic, we choose Bagel as it is a widely recognized and lightweight base model that has superior performance among other options. To adapt UMMs to temporal data, we introduce a fidelity-preserving **Bi**directional **T**ime **S**eries ⇔ **I**mage mappings (**Bi-TSI**), consisting of a TS2I converter and an I2TS converter (Section 3.1). Specifically, the TS2I converter transforms raw time series into a high-fidelity visual representation (TS-image $I$), which is then fed into the backbone model. Within the backbone, the data flow differs by task (the data construction pipeline is described in Section 3.2): (1) **Understanding tasks**: Given an understanding instruction $C_{\text{und}}$ and the TS-image $I$, the *Understanding Model* first generates an understanding CoT $R$, followed by the final understanding answer $o$. (2) **Generation tasks**: The process follows an "understand-then-generate" paradigm. The model first inputs a generation instruction alongside the TS-image $I_{\text{src}}$ into the *Understanding Model* to derive a generation-oriented CoT $R_{\text{gen}}$. This CoT then serves as a conditional guide, and the TS-image $I_{\text{src}}$ is fed again into the *Generation Module*, which synthesizes the target TS-image $I_{\text{tgt}}$. The output TS-image is converted back to numerical time series $o$ via the I2TS converter.

**Training Objectives.** We jointly train the *Understanding Model* and the *Generation Module*. For generation tasks, the generation CoT is produced by the understanding model and is therefore supervised by the understanding loss.

**Understanding Loss (Text).** Given a TS-image $I$ and an instruction $C$, we optimize next-token prediction over a text sequence $y$ (understanding: $y = [R; o]$; generation: $y = R_{\text{gen}}$):

$$\mathcal{L}_{\text{und}} = -\sum_{i=1}^{|y|} \log P_\theta(y_i \mid y_{<i}, I, C). \quad (2)$$

**Generation Loss (Image).** We train the generation module as a diffusion denoiser. Given $I_{\text{tgt}}$, we sample $s$ and add Gaussian noise $\epsilon$ to obtain $I_s$. Here $F_{\text{gen}}(\cdot)$ predicts the injected noise conditioned on $(I_{\text{src}}, R_{\text{gen}})$:

$$\mathcal{L}_{\text{gen}} = \mathbb{E}_{s,\epsilon}\left[\|F_{\text{gen}}(I_s; I_{\text{src}}, R_{\text{gen}}, s) - \epsilon\|_2^2\right]. \quad (3)$$

Ultimately, we minimize a weighted sum of the above losses during training:

$$\mathcal{L} = \lambda_{\text{und}}\,\mathcal{L}_{\text{und}} + \lambda_{\text{gen}}\,\mathcal{L}_{\text{gen}}. \quad (4)$$

### 3.1. Fidelity-Preserving "Time Series ⇔ Image"

To unlock UMMs for time series, we require fidelity-preserving bidirectional mappings that enable near-lossless transformations between time series and TS-image. Therefore, we introduce **Bi-TSI**, which consists of two components: a Time Series-to-Image (TS2I) converter that encodes numerical sequences into a TS-image (Figure 2a) and an Image-to-Time Series (I2TS) converter that decodes a TS-image back to numerical values (Figure 2b).

**Quick Overview of TS2I and I2TS.** Given a multivariate time series $\mathbf{X} \in \mathbb{R}^{T \times N}$ with periodicity $f$, we set the TS-image $I$ to have resolution $H \times W$. (1) TS2I first normalizes $\mathbf{X}$ and folds each variable $\tilde{\mathbf{x}}^{(n)} \in \mathbb{R}^T$ into a periodic grid $\mathbf{S}^{(n)} \in \mathbb{R}^{f \times N_p}$ with $N_p = T/f$. Each grid is then rendered into a band of size $h \times W$, where $h = \lfloor H/N \rfloor$,

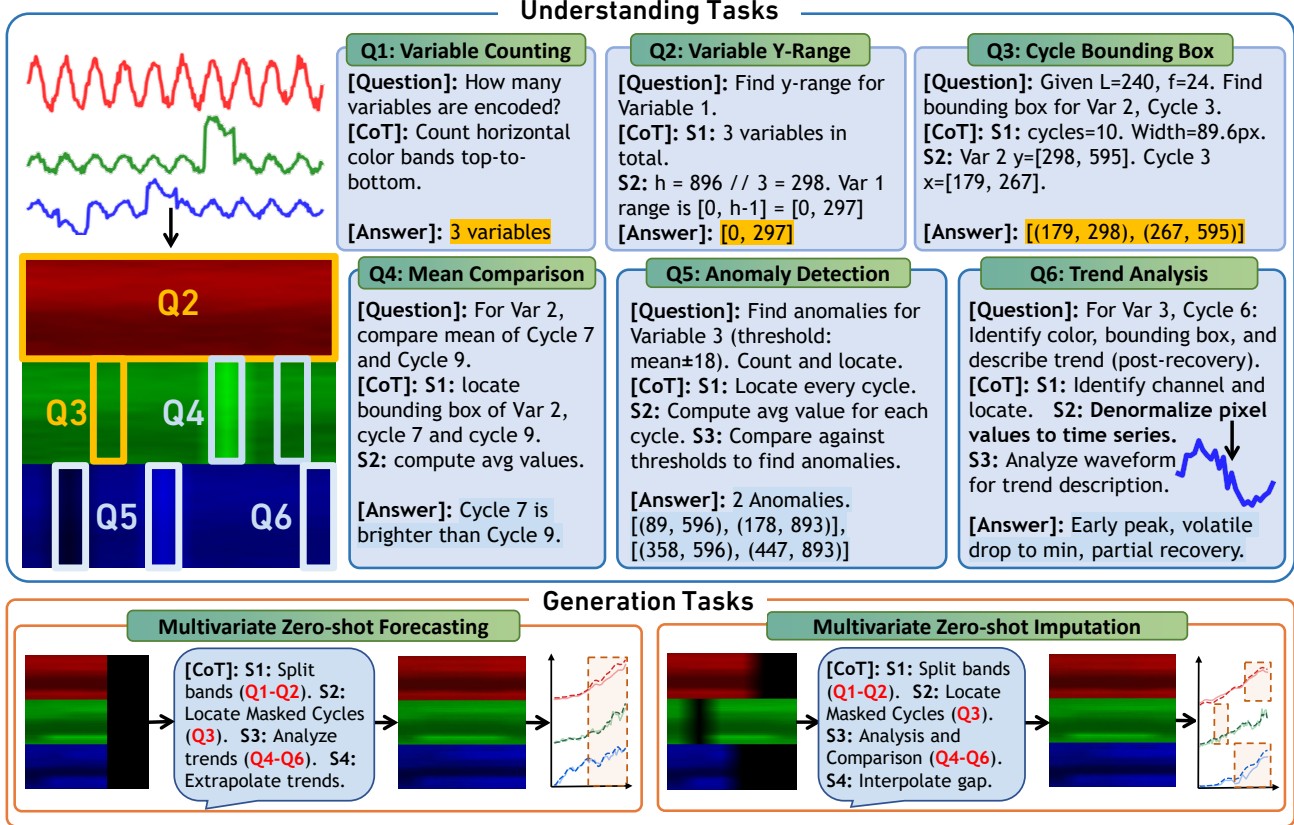

*Figure 4.* Illustrative examples of the proposed **TSUMM-SUITE**, consisting of six time series understanding tasks and two generation tasks. The generation CoT is directly derived from the understanding tasks, explicitly bridging the two task families.

and all bands are stacked vertically to form a TS-image of resolution $H \times W$; a task-specific masking scheme is applied so that the unmasked region provides the observed context while the masked region is completed by the backbone model. (2) I2TS reverses this process by taking the backbone output TS-image, extracting each variable band according to its vertical location, resizing the decoded region back to the $f \times N_p$ grid, unrolling it to the temporal axis, and applying denormalization to recover numerical values. Our conversion pipeline follows the VisionTS++ (Shen et al., 2025), with a step-by-step description provided in Appendix C. In this section, we present two key improvements that make the TS2I/I2TS round-trip mapping reliable for UMMs.

**Robust Fidelity Normalization (RFN).** A key step in TS2I is normalization when projecting values into the image space, but common choices can distort the TS-image. Standard Deviation (Std)-based scaling (Chen et al., 2025b) is sensitive to extreme spikes; a single outlier can compress normal samples into a narrow range, pushing the spike to the boundary. Consequently, the spike geometry may appear saturated in the TS-image (Figure 3a). Meanwhile, Median Absolute Deviation (MAD)-based scaling (Ansari

et al., 2025) fails when many samples share the same value; a near-zero MAD leads to overly aggressive normalization, amplifying minor fluctuations. To address this, RFN combines robust scaling with bounded compression. Given $\mathbf{X} \in \mathbb{R}^{T \times N}$, we compute a per-variable median location $\boldsymbol{\mu} \in \mathbb{R}^N$. For robust scaling $\boldsymbol{\sigma}$, we combine a MAD-based estimate with the standard deviation:

$$\boldsymbol{\sigma} = \alpha \frac{\text{Median}(|\mathbf{X} - \boldsymbol{\mu}|)}{c_{\text{MAD}}} + (1 - \alpha)\,\text{Std}(\mathbf{X})\,. \quad (5)$$

We then apply a smooth bounded mapping via $\tanh$:

$$\mathbf{X}_{\text{norm}} = \tanh\left(\frac{\mathbf{X} - \boldsymbol{\mu}}{\kappa\,\boldsymbol{\sigma}}\right), \quad (6)$$

where $\alpha \in [0, 1]$, $c_{\text{MAD}}$ is the consistency constant, and $\kappa$ controls saturation. See Appendix D for further comparisons of Std-based and MAD-based normalization under two challenging regimes (extreme outliers and step-like signals), and how RFN avoids signal washout and noise amplification.

**Avoiding downsampling via Encoding Capacity Control.** Without explicit constraints on variables or length, VisionTS++ (Shen et al., 2025) maps oversized periodic

grids to the target TS-image resolution, triggering downsampling and loss of temporal details. As shown in Figure 3b, once information is lost at the input stage, even perfect completion fails to recover it, as the backbone cannot restore details removed by the initial mapping. To avoid this failure mode, we make two changes: **(1) capacity constraints to eliminate downsampling** by requiring $H/N \geq f$ and $W \geq L/f$, where $H$ is the available vertical height, $W$ is the horizontal width allocated to the encoded segment, $f$ is the periodicity, $N$ is the number of variables, and $L$ is the total encoded length (including masked portions). These constraints ensure at least one pixel per timestep during rendering, preserving high-fidelity inputs for the backbone model. **(2) higher-resolution TS-images to retain practical capacity**. We use $896 \times 896$ images, providing $16\times$ more area than $224 \times 224$ in VisionTS++, which allows Bi-TSI to encode more variables and longer sequences.

### 3.2. Formulating Generation and Understanding Tasks

We introduce **TSUMM-SUITE**. To leverage understanding for superior generation, we adopt a generation-first pipeline: we first specify generation tasks and then construct understanding samples grounded in them. Detailed case studies can be found in Appendix F.

**Generation Tasks.** We focus on two key time series generation tasks: forecasting and imputation (Figure 4). The pretraining dataset is derived from GIFT-Eval (Aksu et al., 2024). For forecasting, we follow the GIFT-Eval evaluation protocol to adjust the prediction length $P$ based on the series frequency. To ensure the visual loss focuses sufficiently on the completion region, we constrain the context length $L_{ctx}$ to between $P$ and $2P$ for forecasting, and set the masking ratio between $10\%$ and $50\%$ of the total sequence $\mathbf{X}$ for imputation. We construct $40k$ samples for forecasting and $40k$ for imputation. Within each task category, the ratio of univariate, multi-attribute, and multi-node samples is $2 : 1 : 1$. For multivariate samples, the maximum number of input variables in our training set is 21, consistent with the maximum target variables required in the GIFT-Eval testbed.

**Understanding Tasks.** We design six types of understanding tasks tailored to the generation samples (Figure 4). They span two levels: (1) **Layout-level tasks** for locating specific variables and periods, and (2) **Signal-level tasks** for detailed intra-period and inter-period pattern analysis. This hierarchical design compels the model to interpret the TS-image as structured temporal signals rather than superficial textures. Based on the generation samples, we construct $9,409$ QA pairs accompanied by detailed understanding CoTs generated via rules and LLMs (Gemini, 2025). To further enhance temporal reasoning, we also incorporate

the TSR-Suite dataset (Guan et al., 2026), providing $2,339$ CoT-guided temporal reasoning samples to inject essential temporal priors into the understanding model.

**Bridging Generation and Understanding Tasks.** To implement understanding-guided generation, we derive the generation CoT $R_{gen}$ by composing the analytical logic from the understanding tasks (Figure 4). This is feasible because our understanding QAs are constructed on the same generation instances: while layout-level QAs identify the temporal coordinates of variables and periods, signal-level QAs analyze the patterns within these regions. Consequently, the derived $R_{gen}$ integrates these analyses to provide a structured context for the input TS-image $I_{src}$. We structure the training samples as an interleaved sequence:

$$\mathbf{seq} = P_{sys} \oplus I_{src} \oplus C_{gen} \oplus R_{gen} \oplus I_{tgt}, \quad (7)$$

where $P_{sys}$ denotes the system prompt, $C_{gen}$ is the generation instruction, and $I_{tgt}$ is the ground-truth target TS-image. Through this construction, $R_{gen}$ serves as a conditioning context, tightly linking the two task families.

## 4. Experiments

**Implementation.** In our experiments, the understanding model and the generation module are initialized from the pretrained Bagel-7B (Deng et al., 2025). All training data come from the proposed TSUMM-SUITE. Although we constructed $40k$ interleaved sequences for each generation task, we only use $5k$ for training in each task and leave the remaining data for further community exploration. For understanding tasks, we use the full $9,409$ QA pairs with detailed understanding CoT. The model is trained on a node with $8\times$ NVIDIA A100 GPUs. We use a base learning rate of $3 \times 10^{-5}$ with a warm-up phase covering $5\%$ of the total training iterations. All input TS-images have a resolution of $896 \times 896$, resulting in approximately $3,000$ visual tokens per image. In the main comparisons, we use the same checkpoint to evaluate all tasks.

**Evaluation Metrics.** We evaluate TIMEOMNI-VL using standard metrics spanning numerical and textual outputs. For forecasting, we report the normalized Mean Absolute Scaled Error (nMASE) in accordance with common practice on the GIFT-Eval testbed (Aksu et al., 2024); for imputation, we also report nMASE under various masking ratios. For TS-image understanding, scores are normalized to $[0, 1]$ (higher is better) based on task-specific criteria in Appendix E.1. For reasoning tasks, we follow the TSR-Suite benchmark (Guan et al., 2026), reporting Accuracy (ACC) for text-output tasks and Mean Absolute Error (MAE) for sequence-output tasks. All reported results are obtained under zero-shot, out-of-distribution evaluation. Due to the limitations of LLMs in counting (especially for generation

tasks) and their tendency to produce repetitive or garbled outputs (especially for understanding tasks), we compute all subsequent evaluation metrics only on model outputs that yield a valid and extractable answer. This protocol reduces confounding effects from differences in instruction-following abilities across models. "–" indicates that the Success Rate (SR) is below 10%, where the results are omitted due to insufficient statistical reliability, and we therefore do not report them.

## 4.1. Main Results

**Time Series Understanding**

**Setup.** We evaluate six TS-image understanding tasks and find that general-purpose VLMs are not directly applicable without dedicated adaptation. For example, Gemini-2.5-Flash achieves zero accuracy on the signal-level QA5 task; a detailed comparison with two Gemini variants is reported in Table 7 (Appendix E.2). This is expected because our understanding tasks are tailored to the TS-images in TSUMM-SUITE. We therefore conduct a controlled comparison between TIMEOMNI-VL and Bagel-7B across all six tasks to test whether post-training enables the base model to understand our TS-images. **Results.** Figure 5 shows that, while the base model attains zero accuracy on three tasks, TIMEOMNI-VL consistently improves answer accuracy on both layout-level tasks, which evaluate localization of variables and periods, and signal-level tasks, which require value comparison and temporal pattern interpretation. In particular, accuracy on QA1 through QA4 approaches 1.0. These results indicate that post-training substantially strengthens temporal understanding of our TS-images, providing a solid foundation for the subsequent understanding-guided generation.

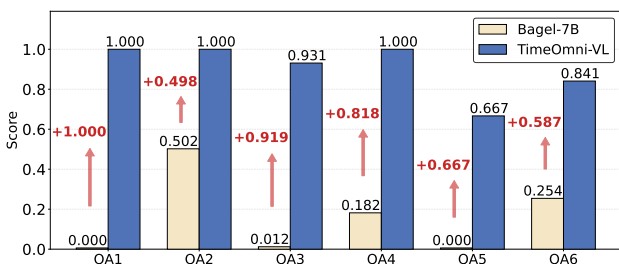

*Figure 5.* Performance on TS-image understanding tasks.

**Time Series Forecasting**

**Setup.** Evaluating the full GIFT-Eval involves over $140k$ sequences, which is impractical for assessing LLMs and UMMs. We adopt a representative subset of 685 instances (419 short-, 137 medium-, and 129 long-term), which is substantially larger than prior TSLM testbeds (Kong et al., 2025). **Results.** Table 1 reports the forecasting results. Among text-output models, Gemini-2.5-Flash (Gemini, 2025) is the only one maintaining reasonable perfor-

mance on long-horizon prediction. Other models (Qwen2.5-Instruct-7B (Yang et al., 2024), Time-R1 (Luo et al., 2025), and TimeOmni-1) fail to reliably forecast at horizons of 480 to 900 steps. This highlights a common bottleneck: deficient counting abilities prevent these models from generating the required sequence length, which precludes quantitative evaluation due to length mismatch. ChatTime (Wang et al., 2025a) is an exception; by mapping each numeric value to a single token, it preserves numerical continuity and improves counting reliability. Even so, these text-based models typically yield nMASE above 1, indicating worse performance than the NAIVE baseline. In contrast, TIMEOMNI-VL and VisionTS-based methods achieve top-tier accuracy. Our base model Bagel-7B fails to forecast without specialized tuning (see Table 23 of Appendix F for failure case). The results show that with dedicated post-training, time series forecasting can be effectively internalized as a capability of UMMs.

**Time Series Imputation**

**Setup.** To ensure zero-shot evaluation, we also use GIFT-Eval and construct a subset of 855 test instances with varying missing ratios: 87 samples with 10%–20% missing, 163 with 20%–30%, 306 with 30%–40%, and 279 with 40%–50%. **Results.** Table 2 reports the imputation results. TIMEOMNI-VL achieves state-of-the-art performance, likely because imputation can leverage both past and future contexts to guide reconstruction, unlike pure forecasting. The untuned Bagel backbone still fails to perform time series-specific task instructions, with representative failure cases provided in Table 24 of Appendix F. Interestingly, simple statistical baselines outperform both the time series-finetuned Moment models (Goswami et al., 2024) and text-only LLM baselines in the imputation task.

**Time Series Reasoning**

*Table 1.* Forecasting performance (nMASE) across different prediction lengths. **Red**: the best, Blue: the 2nd best. "–" denotes SR below 10%; not statistically significant.

| Method | Prediction Length | | |
|---|---|---|---|
| | Short-term | Med-term | Long-term |
| **LLMs** | | | |
| Gemini-2.5-Flash | 1.295 | 1.201 | 1.279 |
| Qwen2.5-Instruct-7B | 1.445 | - | - |
| **Time Series-based Models** | | | |
| ChatTime | 0.983 | 1.439 | 4.164 |
| Time-R1 | 1.162 | - | - |
| TimeOmni-1 | 1.298 | - | - |
| **Image-based Models** | | | |
| VisionTS++ | 0.915 | **0.682** | **0.690** |
| VisionTS | 1.263 | 0.763 | 0.794 |
| Bagel | 16.303 | 17.840 | 16.530 |
| **TIMEOMNI-VL** | **0.878** | 0.816 | 0.784 |

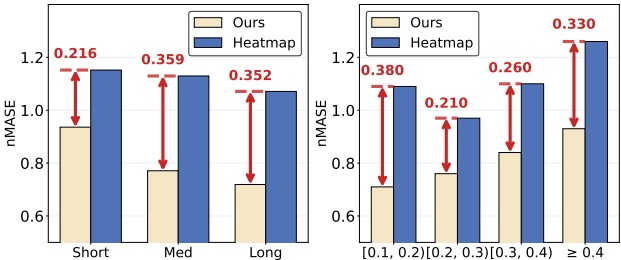

*Figure 6.* Ablation on TS2I strategies. Comparison between our TS2I and the heatmap representation for forecasting (left) and imputation (right). Red arrows indicate the performance gap.

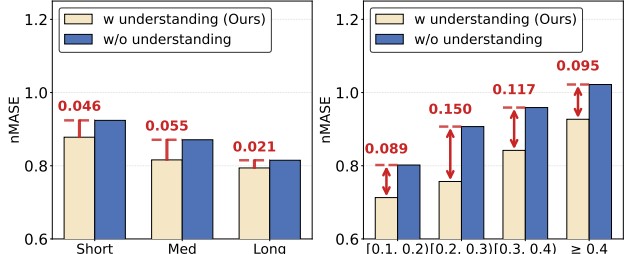

*Figure 8.* Ablation on the understanding model. Comparison between generation-only and understanding-guided generation for forecasting (left) and imputation (right).

**Setup.** To examine whether time series domain knowledge can be effectively injected into UMMs, we follow the out-of-distribution evaluation protocol of TimeOmni-1 (Guan et al., 2026) on text-only reasoning tasks. **Results.** Table 8 in Appendix E.3 reports the reasoning results. Although we do not use reinforcement learning to explicitly enhance the model's reasoning ability, TIMEOMNI-VL achieves top-2 performance on Task 1, Task 2, and Task 4. These results indicate that our post-training successfully incorporates essential time series domain knowledge into UMMs.

*Table 2.* Imputation performance (nMASE) under different masking ratios. **Red**: the best, Blue: the 2nd best. "–" denotes SR below 10%; not statistically significant.

| Method | Masking Ratio | | | |
|---|---|---|---|---|
| | [0.1, 0.2) | [0.2, 0.3) | [0.3, 0.4) | [0.4, 0.5) |
| **LLMs** | | | | |
| Gemini-2.5-Flash | 0.920 | 2.028 | 2.434 | 1.160 |
| Qwen2.5-Instruct-7B | 4.878 | 1.854 | - | - |
| **Statistics Baselines** | | | | |
| Nearest | 0.975 | 0.958 | 1.003 | 0.929 |
| Linear | 0.943 | 0.905 | 0.965 | 0.968 |
| **Time Series-based Models** | | | | |
| Moment-large | 1.220 | 1.400 | 1.630 | 2.100 |
| Moment-base | 1.510 | 1.600 | 1.700 | 2.130 |
| **Image-based Models** | | | | |
| Bagel | 17.411 | 12.239 | 11.849 | 11.032 |
| **TIMEOMNI-VL** | **0.713** | **0.757** | **0.842** | **0.927** |

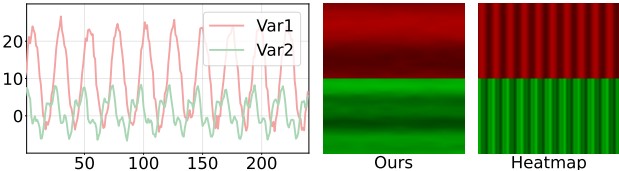

*Figure 7.* Visual comparison of TS-image construction. Original time series (left). Our TS2I strategy (middle), which aligns periodic cycles explicitly. Standard heatmap representation (right).

## 4.2. More Analysis

**Ablation on TS2I Strategies**
**Setup.** We compare our TS2I strategy in Bi-TSI with the widely adopted "time series to heatmap" representation (Ni et al., 2025) (Figure 7). Except for the imaging procedure, all experimental settings are kept identical. We report performance on the generation tasks. **Results.** Figure 6 summarizes the ablation results. Replacing our TS2I with the heatmap representation consistently degrades performance across all tasks; in fact, the heatmap variant yields nMASE worse than the NAIVE baseline on nearly all tasks. This highlights that generation performance is highly sensitive to the choice of TS-image construction strategy. We attribute the degradation to two main factors. (1) **Information loss under limited image resolution.** When the total length (context + prediction) exceeds the TS-image width (896), the heatmap must downsample along the temporal axis, which discards fine-grained information. (2) **Higher modeling difficulty.** Heatmaps require the model to implicitly align periodic patterns across the 2D layout, whereas our TS2I rearranges the series by cycles, making the periodic alignment explicit. We also include a discussion on why we do not use line plots in Appendix E.6.

**Ablation on Understanding Model**
**Setup.** To verify whether understanding can facilitate generation, we freeze the understanding model during training and disable CoT generation during inference. **Results.** Figure 8 summarizes the ablation results. Without CoT as context, generation performance drops consistently across all cases, yielding an average 8.2% increase in nMASE. This suggests that the shared self-attention in our backbone model enables effective interaction between the understanding model and the generation module, allowing the generation module to leverage the semantics provided by the understanding model and consequently produce more controllable time series generations.

**Case Studies**
Detailed case studies across all tasks (six understanding, two generation) are provided in Appendix F. Additionally, we present two representative failure cases of the base model in

Table 23 and Table 24. These comparisons further demonstrate that our post-training internalizes time series understanding and generation as inherent capabilities of UMMs.

# 5. Conclusion

We introduced TIMEOMNI-VL, a vision-centric framework that unifies temporal understanding and generation. We first develop Bi-TSI, a fidelity-oriented mapping that ensures near-lossless time series-to-image conversion. Building on this, we introduce TSUMM-SUITE, a benchmark comprising comprehensive understanding tasks that advance the model from basic periodic localization to complex pattern analytics, alongside downstream generation tasks. Through an understanding-guided generation mechanism formulated as a CoT-conditioned process, TIMEOMNI-VL links semantic understanding to high-fidelity generation. Experimental results demonstrate that TIMEOMNI-VL performs strongly on both understanding and generation, providing a new perspective on vision-centric unified time series modeling.

# Impact Statement

This paper presents work whose goal is to advance the field of machine learning and time series analytics. There are many potential societal consequences of our work, none of which we feel must be specifically highlighted here.

# Acknowledgment

S. Pan was partially supported by the Australian Research Council (ARC) under grants FT210100097 and DP240101547, and the CSIRO–National Science Foundation (US) AI Research Collaboration Program. This work was also supported by the NVIDIA Academic Grant in Higher Education and Developer Program.

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

# A. Dataset Details

## A.1. Data Statistics

This section reports the quantitative statistics of the proposed TSUMM-SUITE. As summarized in Table 3, TSUMM-SUITE is constructed for post-training to equip our model with unified time series understanding and generation capabilities. It comprises two generation tasks (forecasting and imputation), one TS-image understanding task suite, and one reasoning task set. For generation, we provide 40,000 training instances for each of forecasting and imputation, together with testbeds of 685 and 855 instances, respectively. For understanding, we include 9,409 training QA pairs tailored to our TS-image representation and a 685-instance test set for evaluation. For reasoning, we incorporate the TSR-Suite (Guan et al., 2026) split, with 2,339 training and 2,448 test samples, which serves as high-quality instruction tuning data to improve generalizable temporal reasoning.

*Table 3.* Detailed quantitative statistics for the four time series tasks in TSUMM-SUITE across training sets and testbeds.

|  | Forecasting | Imputation | Understanding | Reasoning |
|---|---|---|---|---|
| **Training Set** | 40,000 | 40,000 | 9,409 | 2,339 |
| **Testbed** | 685 | 855 | 685 | 2,448 |

## A.2. Statistics on Sequence Length and Token Budget

In this section, we report the actual sequence lengths used in TSUMM-SUITE in Table 4 and the corresponding token budgets computed with the tokenizer of our base model Bagel in Table 5.

As shown in Table 4, TSUMM-SUITE covers a wide range of temporal scales. Forecasting, imputation, and understanding involve long-range dependencies, with a maximum length of 2,592 and an average of about 950 time points.

*Table 4.* Maximum, minimum, and average time series lengths across four tasks.

|  | Forecasting | Imputation | Understanding | Reasoning |
|---|---|---|---|---|
| **MAX length** | 2,592 | 2,592 | 2,592 | 792 |
| **MIN length** | 8 | 8 | 8 | 10 |
| **AVG length** | 957 | 957 | 947 | 109 |

Table 5 reports the average token usage for the time series input ($\mathbf{X}$) and textual context ($C$) across tasks. For forecasting and imputation, inputs are predominantly visual, with an average of 7,236 image tokens and about 130 context tokens. For understanding, the textual component increases to 479 context tokens on average, alongside 4,096 visual tokens. Finally, the reasoning task uses 860 time series tokens and 246 context tokens on average.

*Table 5.* Average token budgets computed using the tokenizer of our base model Bagel (Deng et al., 2025).

|  | Forecasting | Imputation | Understanding | Reasoning |
|---|---|---|---|---|
| **AVG tokens of time series X** | 7,236 | 7,236 | 4,096 | 860 |
| **AVG tokens of context $C$** | 116 | 130 | 479 | 246 |
| **AVG total tokens** | 7,352 | 7,366 | 4,575 | 1,106 |

## B. Prompts Used in This Paper

### B.1. Prompt for Gemini to Generate Time Series Pattern Analyses

---

**Prompt for Gemini to Generate Pattern Analyses**

You will be given a single-cycle time series for variable {`var_idx`} (cycle {`cycle_idx`}, {`color`} channel).

Time series (length={`T`}):
`[21.6, 21.7, 31.7,...]`

Helpful stats (do not ignore the raw series):
- min={`arr.min():.3f`} at $t = \{$`valley_i`$\}$, max={`arr.max():.3f`} at $t = \{$`peak_i`$\}$
- start={`arr[0]:.3f`}, end={`arr[-1]:.3f`}

Write a concise 2–3 sentence description of the trend/shape using only what is evident from the series.

Guidelines:
- Describe the overall trend direction (increasing/decreasing/flat). If the behavior changes over time, you may describe it in 2–3 phases (e.g., early/mid/late), but do not force segmentation if the series is stable.
- Mention whether the series fluctuates and whether fluctuations are small/moderate/large (relative to its range).
- Mention any notable peak, valley, or plateau, and roughly when it occurs (early/mid/late or with an index $t = \ldots$).
- If there is no clear peak/valley/plateau, explicitly say so.

Return a single paragraph. Do NOT invent events not supported by the numbers.

---

### B.2. System Prompt for Training and Evaluation

This section presents the system prompts used for training and evaluation (Section 4). We categorize them into two types: understanding task system prompts and generation task system prompts.

---

**System Prompt for Time Series Understanding Tasks.**

You should first think about the reasoning process in the mind and then provide the user with the answer.
The reasoning process is enclosed within `<think>` `</think>` tags, i.e. `<think>` reasoning process here `</think>` answer here

---

**System Prompt for Time Series Generation Tasks.**

You should first think about the planning process in the mind and then generate the image.
The planning process is enclosed within `<think>` `</think>` tags, i.e. `<think>` planning process here `</think>` image here

---

## C. Details of the TS2I and I2TS Process

In this section, we provide a detailed description of the bidirectional mappings between time series and images utilized throughout TIMEOMNI-VL. Our goal is a fidelity-preserving Time Series ⇔ Image transformation that is as close to lossless as possible. This requirement is crucial because the TS-image is fed into the UMM backbone as the model input. If the TS2I conversion discards numerical information, the backbone cannot recover it, and the entire vision-centric pipeline would fail to produce high-fidelity time series outputs. Likewise, the image generated by the backbone must be decoded back to a numerical sequence without losing the information contained in the output image. Therefore, we design TS2I and I2TS as a deterministic round-trip mapping and treat it as near-lossless in practice, with residual errors primarily arising from spatial interpolation and finite numerical precision.

## C.1. Time Series to Image (TS2I) Converter

**Periodicity-based segmentation.** Given a multivariate time series $\mathbf{X} \in \mathbb{R}^{T \times N}$ with periodicity $f \in \mathbb{Z}^+$, we adopt a periodicity-consistent setting in our experiments, where both the context length and the prediction horizon are integer multiples of $f$. If the available length is not an exact multiple of $f$, we truncate it to the nearest valid length. Consequently, $T$ is divisible by $f$ and the series can be decomposed into $C = T/f$ periodic blocks without padding. Prior to periodic segmentation, $\mathbf{X}$ is normalized using robust fidelity normalization (RFN) in Section 3.1 to ensure numerically stable and geometry-consistent rendering.

**Rearrangement into a periodic grid.** For each variable $n$, let $\tilde{\mathbf{x}}^{(n)} \in \mathbb{R}^T$ denote the normalized sequence after applying RFN, where $\tilde{(\cdot)}$ indicates values in the normalized space used for image rendering. We fold the normalized sequence $\tilde{\mathbf{x}}^{(n)}$ into a $f \times C$ matrix $\mathbf{S}^{(n)} \in \mathbb{R}^{f \times C}$, where $C = T/f$:

$$\mathbf{S}_{i,j}^{(n)} = \tilde{\mathbf{x}}_{jf+i}^{(n)}, \qquad i = 0, \ldots, f-1, \ j = 0, \ldots, C-1. \tag{8}$$

Here, the row index $i$ corresponds to the intra-period position, while the column index $j$ indexes successive periods. This construction maps intra-period structure to vertical locality and inter-period evolution to horizontal progression.

**Rendering.** Given $\mathbf{S}^{(n)} \in \mathbb{R}^{f \times C}$, the rendering step upsamples the periodic grid into the image coordinate space. Specifically, we allocate each variable a vertical band of height $h = \lfloor H/N \rfloor$ and resize $\mathbf{S}^{(n)}$ to $h \times W_{\text{in}}$, where $W_{\text{in}}$ denotes the width of the unmasked region and the remaining width $W_{\text{out}} = W - W_{\text{in}}$ is masked. For forecasting, the mask occupies the right side so the model completes future periods from left to right; for imputation, masked regions can be placed at arbitrary locations within the TS-image.

**Supporting multivariate inputs via band stacking and color assignment.** For the multivariate time series input $\mathbf{X}$, TS2I renders each variable into one band and stacks the $N$ bands along the vertical axis to construct the complete TS-image, whose overall resolution is $H \times W$ with the visible context occupying the left width $W_{\text{in}}$. To distinguish different variables within a single image, we follow the setting of VisionTS++ (Shen et al., 2025) and assign each band a RGB color, while enforcing that adjacent bands do not share the same color. This simple color assignment preserves the band geometry and helps the backbone model separate variable-specific patterns in the visual space.

## C.2. Image to Time Series (I2TS) Converter

**Recovering the completed region and inverse rearrangement.** Given the output TS-image $\hat{I} \in \mathbb{R}^{H \times W}$, I2TS decodes numerical values from the completed region. We first recover each variable band according to its vertical location using the same band height $h = \lfloor H/N \rfloor$ as in TS2I. For variable $n$, we crop its band from $\hat{I}$ and resize the decoded region back to the periodic grid resolution $f \times C$, yielding $\hat{\mathbf{S}}^{(n)} \in \mathbb{R}^{f \times C}$. Finally, we invert the TS2I folding step to obtain the normalized sequence $\hat{\mathbf{x}}^{(n)} \in \mathbb{R}^T$:

$$\hat{\mathbf{x}}_{jf+i}^{(n)} = \hat{\mathbf{S}}_{i,j}^{(n)}, \qquad i = 0, \ldots, f-1, \ j = 0, \ldots, C-1. \tag{9}$$

Concatenating all variables gives the normalized multivariate sequence $\hat{\mathbf{U}} \in \mathbb{R}^{T \times N}$ for the decoded region.

**Inverse normalization and value restoration.** I2TS applies the exact inverse of the RFN mapping defined in Equation 6. Let $\hat{\mathbf{U}}$ denote the decoded values in the normalized space. We first apply inverse hyperbolic tangent:

$$\hat{\mathbf{Z}} = \kappa \operatorname{arctanh}\left(\hat{\mathbf{U}}\right), \tag{10}$$

where values in $\hat{\mathbf{U}}$ are implicitly clamped within the valid domain $(-1, 1)$ for numerical stability. Finally, we restore the original numerical scale using the per-variable statistics $(\boldsymbol{\mu}, \boldsymbol{\sigma})$ recorded during the encoding stage:

$$\hat{\mathbf{X}} = \hat{\mathbf{Z}} \odot \boldsymbol{\sigma} + \boldsymbol{\mu}. \tag{11}$$

In summary, TS2I and I2TS form a deterministic round-trip mapping that is near-lossless in practice. Any residual reconstruction error mainly comes from spatial interpolation introduced in rendering and resizing, rather than from stochasticity in the transformation itself.

## D. Comparison of Different Normalization Strategies

In this section, we provide an intuitive explanation of why existing normalization methods (for example, standard deviation (Std)-based (Chen et al., 2025b) and median absolute deviation (MAD)-based normalization (Ansari et al., 2025)) fall short and how our robust fidelity normalization (RFN) addresses these issues. We focus on two extreme yet common regimes: signals with extreme outliers and signals with step-like patterns.

### D.1. Case I: Extreme Outliers

**Scenario.** Assume a clean informative signal (e.g., a sine wave) contaminated by a single, massive outlier with amplitude $\Delta$. This creates a single abrupt spike in the signal. The standard deviation $\sigma$ is highly sensitive to extreme values. A single massive spike causes $\sigma$ to grow with the outlier size ($\sigma \approx \Delta/\sqrt{T}$). Consequently, for the normal part of the signal $x_t$, the normalized value $\hat{x}_t$ collapses:

$$\hat{x}_t \approx \frac{x_t}{\Delta/\sqrt{T}} \xrightarrow{\Delta \to \infty} 0. \tag{12}$$

When applied to TS2I conversion, the informative signal is compressed toward zero. As a result, the outlier is mapped to a single bright pixel in the TS-image, while the underlying temporal patterns collapse into a nearly uniform dark background. The vision backbone consequently focuses almost exclusively on the outlier.

**RFN Solution.** RFN uses the MAD, which ignores the single outlier, keeping the denominator stable. The outlier is smoothly saturated by the bounded `tanh` function, preserving the visibility of the main signal.

### D.2. Case II: Signals with Step-like Patterns

**Scenario.** Consider a "step function" or a signal $\mathbf{x}$ that stays constant for a long period. In these flat regions, the value at time $t$ can be expressed as $x_t = c + \eta_t$, where $c$ is a constant and $\eta_t$ represents microscopic noise. For a signal that is constant for more than half of its length, the MAD is mathematically zero. This leads to division by zero, causing the microscopic noise to be amplified to massive magnitudes:

$$\hat{x}_t \approx \frac{\eta_t}{0} \to \infty. \tag{13}$$

When applied to TS2I conversion, the normalization artificially amplifies negligible sensor noise into high-amplitude pixel-level fluctuations. As a result, the TS2I becomes dominated by high-contrast artifacts, falsely suggesting violent temporal variability in the input signal.

**RFN Solution.** RFN prevents this collapse by incorporating the standard deviation as a regularizing term. Even if MAD is zero, the standard deviation of a step function remains non-zero, providing a "safety floor":

$$\sigma_{\mathrm{RFN}} = \alpha \cdot \underbrace{\mathrm{MAD}(\mathbf{x})}_{\approx 0} + (1 - \alpha) \underbrace{\mathrm{Std}(\mathbf{x})}_{>0}, \tag{14}$$

where $\sigma_{\mathrm{RFN}}$ denotes the robust scaling factor used by RFN. This ensures that the resulting image correctly depicts flat regions with clear transitions.

Table 6 summarizes the behavior of each normalization strategy across representative regimes. RFN is the only method that consistently performs ideal TS2I conversion, remaining effective in both outlier-dominated signals and step-like signals with extended flat regions.

*Table 6.* Qualitative behavior of different normalization methods under representative challenging regimes. Ideal indicates faithful visual preservation of the underlying signal structure.

| Regime | Std-based | MAD-based | RFN (Ours) |
|---|---|---|---|
| Gaussian signal | Ideal | Ideal | Ideal |
| Heavy outliers | Signal washout | Ideal | Ideal |
| Step / flat | Ideal | Noise amplification | Ideal |

# E. Additional Experimental Results

## E.1. The Scoring Criteria for Understanding Tasks

To ensure a rigorous evaluation of the model's ability to interpret TS-images, we design specific scoring metrics for each understanding task. All scores are normalized to the range $[0, 1]$. The detailed criteria are defined as follows:

- **Understanding QA1: Variable Counting.** We utilize exact match (EM). The score is $1$ if the predicted integer representing the number of variables exactly matches the ground truth, and $0$ otherwise.

- **Understanding QA2: Variable Y-Range.** We evaluate the model's ability to localize variables vertically using the intersection over union (IoU) metric. For each variable, its vertical span is represented as a rectangular region covering the full width of the segment. Let $B_{pred}$ and $B_{gt}$ denote the predicted and ground-truth bounding boxes, respectively. The score is calculated as:

$$\text{Score} = \text{IoU}(B_{pred}, B_{gt}) = \frac{\text{Area}(B_{pred} \cap B_{gt})}{\text{Area}(B_{pred} \cup B_{gt})}. \tag{15}$$

- **Understanding QA3: Cycle Bounding Box.** Similarly, we utilize bounding box IoU. The model outputs the specific coordinates $[(x_1, y_1), (x_2, y_2)]$ for a cycle. The score is the IoU between the predicted bounding box $B_{pred}$ and the ground-truth box $B_{gt}$, calculated using the same formula as QA2.

- **Understanding QA4: Mean Comparison.** We utilize EM. The task requires identifying which of two specific cycles has a higher mean value. The score is $1$ if the predicted cycle index exactly matches the ground-truth index (e.g., correctly selecting "Cycle 7" over "Cycle 9"), and $0$ otherwise.

- **Understanding QA5: Anomaly Detection.** We utilize weighted accuracy. We parse the output to extract three key count statistics: the total count of anomalous cycles, the count of bright anomalies, and the count of dark anomalies. The final score is the average of the match results for these three components (each contributing $1/3$). For example, if the ground truth is "2 anomalous cycles (1 bright, 1 dark)" and the model correctly predicts all three counts, the score is $1$; if it correctly predicts the total and bright counts but misses the dark count, the score is $2/3$.

- **Understanding QA6: Trend Analysis.** We utilize a composite score consisting of three equally weighted sub-components ($1/3$ each):

  1. **Color Consistency:** We use EM. The score is $1$ if the predicted color channel (e.g., "Blue") exactly matches the ground truth, and $0$ otherwise.
  2. **Localization Accuracy:** We use bounding box IoU between the predicted bounding box and the ground-truth box (between 0 and 1).
  3. **Trend Description Quality:** We use BERTScore (Zhang* et al., 2020) to measure the semantic similarity between the generated textual description and the ground-truth analysis.

  The final score is the arithmetic mean of these three sub-scores: $\text{Score} = \frac{1}{3}(\text{EM}_{\text{color}} + \text{IoU}_{\text{bbox}} + \text{BERTScore}_{\text{text}})$.

## E.2. Results of Understanding Tasks

*Table 7.* Performance on Understanding Tasks. The table reports scores for layout-level tasks (QA1–3) and signal-level tasks (QA4–6).

| Method | Layout Tasks | | | Signal Tasks | | |
|---|---|---|---|---|---|---|
| | QA1 | QA2 | QA3 | QA4 | QA5 | QA6 |
| **Proprietary VLMs** | | | | | | |
| Gemini-2.5-Flash | 0.540 | 0.640 | 0.004 | 0.535 | 0 | 0.342 |
| Gemini-2.0-Flash | 0.230 | 0.290 | 0.261 | 0.279 | 0 | 0.220 |
| **Base Model** | | | | | | |
| Bagel | 0 | 0.502 | 0.012 | 0.182 | 0 | 0.254 |
| **TIMEOMNI-VL** | 1 | 1 | 0.931 | 1 | 0.667 | 0.841 |

## E.3. Results of Reasoning Tasks

*Table 8.* Performance on Reasoning Tasks. The default metric is ACC, except for Task 3 where MAE is used. **Red**: the best, Blue: the 2nd best. "–" denotes SR below 10%; not statistically significant.

| Method | Perception | | Extrapolation | Decision Making |
|---|---|---|---|---|
| | Task1 | Task2 | Task3↓ | Task4 |
| **LLMs** | | | | |
| Gemini-2.5-Flash | 77.5 | 25.9 | 170.78 | 36.6 |
| Qwen2.5-Instruct-7B | 42.8 | 26.3 | 146.12 | 24.9 |
| **TSLMs** | | | | |
| Time-MQA-8B (Kong et al., 2025) | 25.1 | 31.2 | - | 11.6 |
| ChatTS (Xie et al., 2025) | 39.2 | 18.6 | - | 11.1 |
| ITFormer (wang et al., 2025) | 47.5 | 14.6 | 230.04 | 41.7 |
| Time-R1 (Luo et al., 2025) | 34.0 | 31.4 | 160.47 | 32.2 |
| TimeOmni-1 (Guan et al., 2026) | **87.7** | **64.0** | **145.53** | 58.9 |
| **TIMEOMNI-VL** | 84.0 | 61.3 | 163.79 | **61.4** |

## E.4. Additional Comparison Results on Generation Tasks

### E.4.1. CONTROLLED COMPARISON WITH VISIONTS++ ON GIFT-EVAL

To provide a controlled comparison with image-based generation models, we retrain VisionTS++ on the same $5k$ generation training samples used by TIMEOMNI-VL. Table 9 reports the results on the GIFT-Eval subsets. Under the same training-data budget, TIMEOMNI-VL achieves better average performance on both forecasting and imputation, while VisionTS++ remains competitive on some individual horizons and masking ratios.

*Table 9.* Controlled comparison with VisionTS++ on GIFT-Eval subsets. We report nMASE for forecasting and imputation. VisionTS++ is retrained on the same $5k$ training samples as TIMEOMNI-VL.

| | Forecasting (nMASE) | | | | Imputation (nMASE) | | | | |
|---|---|---|---|---|---|---|---|---|---|
| | Short | Med | Long | AVG | [0.1,0.2) | [0.2,0.3) | [0.3,0.4) | [0.4,0.5] | AVG |
| VisionTS++ (retrained on our training set) | 1.106 | **0.798** | 0.797 | 0.900 | 0.729 | 0.818 | 0.853 | **0.908** | 0.827 |
| **TIMEOMNI-VL** | **0.878** | 0.816 | **0.784** | **0.826** | **0.713** | **0.757** | **0.841** | 0.927 | **0.810** |

### E.4.2. ZERO-SHOT GENERALIZATION ON TIME BENCHMARK SUBSETS

We further evaluate TIMEOMNI-VL on TIME (Qiao et al., 2026) benchmark subsets under a strict zero-shot protocol. For forecasting, VisionTS++ is evaluated with both its original checkpoint and an aligned version retrained on the same $5k$ training samples as TIMEOMNI-VL. For imputation, we report the aligned VisionTS++ result together with statistical and time series baselines. The results in Tables 10 and 11 show that TIMEOMNI-VL remains effective beyond the GIFT-Eval testbed.

*Table 10.* Results on TIME benchmark subsets for forecasting across short-, medium-, and long-term horizons. We report nMASE under zero-shot evaluation. VisionTS++ (aligned) denotes retraining on the same $5k$ training samples as TIMEOMNI-VL.

| | Forecasting (nMASE) | | | |
|---|---|---|---|---|
| Model | Short | Med | Long | AVG |
| ChatTime | 0.994 | 1.158 | 2.289 | 1.480 |
| VisionTS | 0.933 | 0.870 | 0.840 | 0.881 |
| VisionTS++ | 0.830 | 0.921 | **0.836** | 0.862 |
| VisionTS++ (aligned) | 0.833 | 0.926 | 1.008 | 0.922 |
| **TIMEOMNI-VL** | **0.815** | **0.832** | 0.926 | **0.858** |

*Table 11.* Results on TIME benchmark subsets for imputation. We report zero-shot nMASE. VisionTS++ (aligned) denotes retraining on the same $5k$ samples as TIMEOMNI-VL.

| | Imputation (nMASE) | | | | |
|---|---|---|---|---|---|
| Model | [0.1,0.2) | [0.2,0.3) | [0.3,0.4) | [0.4,0.5] | AVG |
| Moment-large | 0.942 | 1.023 | 1.150 | 1.195 | 1.078 |
| Moment-base | 1.103 | 1.104 | 1.213 | 1.267 | 1.172 |
| Nearest | 1.082 | 0.981 | 0.968 | 1.018 | 1.012 |
| Linear | 0.998 | 0.936 | 0.933 | 0.985 | 0.963 |
| VisionTS++ (aligned) | 0.962 | 0.956 | 1.063 | 1.164 | 1.036 |
| **TIMEOMNI-VL** | **0.863** | **0.688** | **0.725** | **0.861** | **0.784** |

## E.5. Additional Analysis of TS-image Construction

### E.5.1. IMPACT OF TS-IMAGE RESOLUTION

We evaluate TIMEOMNI-VL under three TS-image resolutions to analyze the trade-off between information preservation and computational cost. Table 12 shows that increasing the resolution consistently improves generation quality, but also increases both training and inference time per image.

*Table 12.* Impact of TS-image resolution on performance and efficiency. We report nMASE for forecasting and imputation.

| | Efficiency | | Forecasting (nMASE) | | | | Imputation (nMASE) | | | | |
|---|---|---|---|---|---|---|---|---|---|---|---|
| Resolution | Training / Image | Inference / Image | Short | Med | Long | AVG | [0.1, 0.2) | [0.2, 0.3) | [0.3, 0.4) | [0.4, 0.5] | AVG |
| 224×224 | 44 ms | 11s | 0.862 | 0.879 | 0.811 | 0.851 | 0.761 | 0.947 | 0.924 | **0.864** | 0.874 |
| 448×448 | 149 ms | 24s | 0.842 | 0.807 | 0.780 | 0.810 | 0.686 | 0.856 | 0.902 | 0.910 | 0.838 |
| 896×896 | 651 ms | 50s | **0.814** | **0.769** | **0.767** | **0.783** | **0.599** | **0.800** | **0.880** | 0.911 | **0.798** |

### E.5.2. EFFECT OF SINGLE-COLOR RENDERING

To evaluate the role of multi-color cues, we render all variables in a single color and evaluate both direct inference and fine-tuning under the single-color setting. Table 13 shows that the original multi-color design remains more effective.

*Table 13.* Effect of single-color TS-image rendering compared with the original multi-color setting. We report nMASE for forecasting and imputation, and average score for understanding.

| | Forecasting (nMASE) | | | | Imputation (nMASE) | | | | | Understanding |
|---|---|---|---|---|---|---|---|---|---|---|
| | Short | Med | Long | AVG | [0.1,0.2) | [0.2,0.3) | [0.3,0.4) | [0.4,0.5] | AVG | AVG |
| Inference-only (single color) | 1.023 | 0.978 | 0.865 | 0.955 | 0.710 | 0.925 | 0.938 | 0.972 | 0.886 | 0.707 |
| Fine-tuning (single color) | 0.859 | 0.798 | 0.792 | 0.816 | 0.627 | 0.857 | 0.891 | 0.922 | 0.824 | 0.893 |
| **TIMEOMNI-VL (multi-color)** | **0.814** | **0.769** | **0.767** | **0.783** | **0.599** | **0.800** | **0.880** | **0.911** | **0.798** | **0.908** |

### E.5.3. ROBUSTNESS TO VARIABLE-ORDER SHUFFLING

To examine whether the model is overly sensitive to the vertical order of variables in a multivariate TS-image, we randomly sample 200 multivariate test samples for forecasting and 200 for imputation, then construct three shuffled evaluation sets by permuting the variable order. As shown in Table 14, the model remains reasonably robust under variable-order shuffling.

*Table 14.* Robustness to shuffling the order of multiple time series within a single TS-image. We report nMASE, and the last row shows the mean and 95% confidence interval (CI).

| | Forecasting | | | Imputation | | | |
|---|---|---|---|---|---|---|---|
| | Short | Med | Long | [0.1,0.2) | [0.2,0.3) | [0.3,0.4) | [0.4,0.5] |
| Original Order | 0.814 | 0.769 | 0.767 | 0.599 | 0.800 | 0.880 | 0.911 |
| Shuffle Group 1 | 0.795 | 0.783 | 0.822 | 0.671 | 0.944 | 0.950 | 0.998 |
| Shuffle Group 2 | 0.818 | 0.804 | 0.824 | 0.672 | 0.927 | 0.950 | 0.947 |
| Shuffle Group 3 | 0.818 | 0.819 | 0.807 | 0.582 | 0.888 | 0.909 | 0.936 |
| **Mean ± 95% CI** | $0.811 \pm 0.018$ | $0.794 \pm 0.035$ | $0.805 \pm 0.042$ | $0.631 \pm 0.075$ | $0.890 \pm 0.102$ | $0.922 \pm 0.055$ | $0.948 \pm 0.058$ |

## E.6. Discussion on Line Plot Representations

We exclude line plots due to four practical limitations. (1) Information sparsity. Most pixels correspond to background, while the signal is confined to thin strokes, which limits representational capacity. (2) Variable overlap. In multivariate settings, intersecting curves create ambiguity, making it difficult to uniquely identify and disentangle variables. (3) Misaligned attention. General-purpose vision-language models (VLMs) and UMMs tend to focus on textual labels and legends rather than the fine geometry of thin lines (Zhou et al., 2025). (4) Decoding complexity. Recovering precise values from rendered curves is an ill-posed inverse problem that is sensitive to stroke width, aliasing, and line overlap, leading to unstable decoding.

# F. Case Study

## F.1. Comprehensive Task Demonstrations of TSUMM-SUITE

In this section, we provide detailed case studies across the six understanding tasks (Tables 15 to 20) and two generation tasks (Tables 21 and 22) within the TSUMM-SUITE benchmark.

*Table 15.* Example of Understanding Task 1: Variable Counting.

---

**Understanding Task 1: Variable Counting**

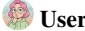 **User**    You are given a 896 × 896 image that encodes one or more time series variables as horizontal bands stacked from top to bottom. How many variables (time series channels) are encoded in this image?

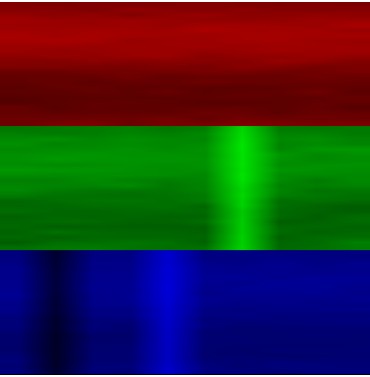

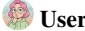 **TIMEOMNI-VL**
```
<think>
```
Each continuous horizontal color band corresponds to one time series channel. Counting the bands from top to bottom gives 3.
```
</think>
Answer:
```
The image encodes 3 variables (time series channels).

---

*Table 16.* Example of Understanding Task 2: Variable Y-Range.

---

**Understanding Task 2: Variable Y-Range**

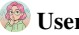 **User**

You are given a 896 × 896 image that encodes one or more time series variables as horizontal bands stacked from top to bottom.
- Variables are indexed from 1 (top band) to nvars (bottom band).
- First, determine nvars by counting the number of horizontal bands in the image.
- Then compute the per-variable height using integer (floor) division: `image_size_per_var = max(1, image_size // max(1, nvars))`
Using y=0 at the top row and inclusive pixel indices, what is the y-range [y_start, y_end] for variable 1?

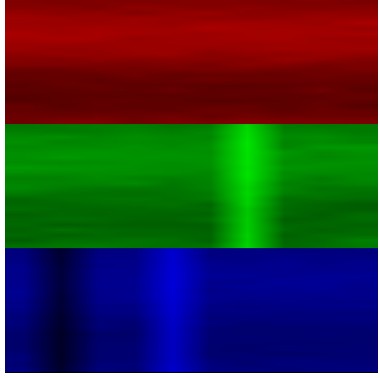

**TIMEOMNI-VL**

```
<think>
```
**1) Visual Analysis.**
Scanning the image reveals 3 distinct horizontal bands. Thus, $n_{vars} = 3$.
**2) Height Calculation.**
Image height $H = 896$.
Per-variable height $h = \lfloor 896/\max(1,3) \rfloor = 298$ pixels.
**3) Range Derivation.**
Target: **Variable 1** (Topmost, internal index $i = 0$).
- $y_{start} = i \times h = 0 \times 298 = 0$.
- $y_{end} = (i+1) \times h - 1 = 1 \times 298 - 1 = 297$.
```
</think>
```

```
Answer:
```
Variable 1 occupies y-range: [0, 297].

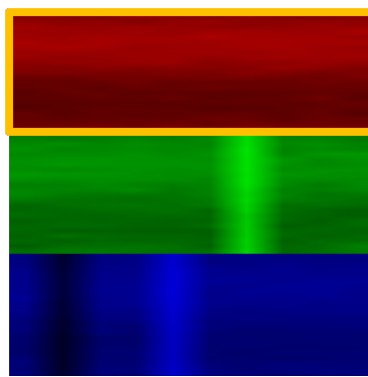

*Table 17.* Example of Understanding Task 3: Cycle Bounding Box.

**Understanding Task 3: Cycle Bounding Box**

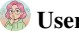 **User**

You are given a $896 \times 896$ image that encodes one or more time series variables as horizontal bands stacked from top to bottom.
Given metadata: $L = 240$, periodicity = 24.
Each cycle corresponds to a vertical strip in the image. Questions:
(1) How many pixels wide is one cycle?
(2) For variable 2 and cycle 3, give the bounding box [(x1, y1), (x2, y2)] of that cycle region.

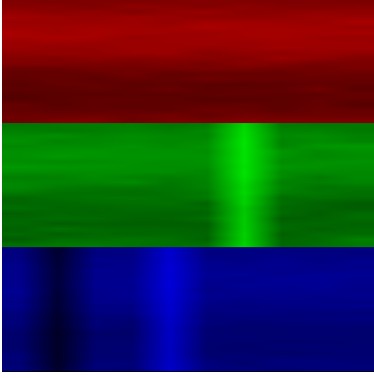

**TIMEOMNI-VL**

```
<think>
```
**1) Geometry & Layout.**
Image size $896 \times 896$. Scanning bands $\rightarrow n_{\text{vars}} = 3$. Band height $h = \lfloor 896/3 \rfloor = 298$.
**2) Cycle Dimensions.**
Metadata: $L = 240$, Periodicity $P = 24$. Total cycles $N = \lceil L/P \rceil = \lceil 240/24 \rceil = 10$. Width per cycle $w = 896/10 = 89.6$ pixels.
**3) Coordinates Calculation.**
Target: **Variable 2**, **Cycle 3**.

- Y-range: $[(2-1)h, 2h-1] = [298, 595]$.

- X-range: $[\lfloor (3-1)w \rfloor, \lfloor 3w \rfloor - 1]$.

- $x_1 = \lfloor 2 \times 89.6 \rfloor = 179$.

- $x_2 = \lfloor 3 \times 89.6 \rfloor - 1 = 267$.

```
</think>
```

```
Answer:
```
(1) Each cycle is 89.6 pixels wide.
(2) The bounding box for variable 2, cycle 3 is: [(179, 298), (267, 595)].

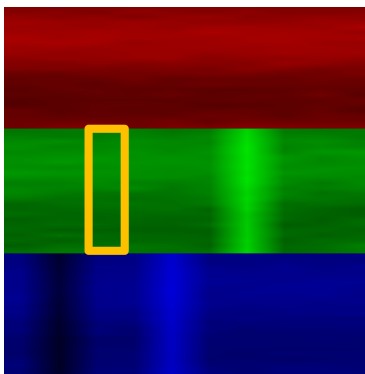

*Table 18.* Example of Understanding Task 4: Mean Comparison.

**Understanding Task 4: Mean Comparison**

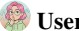 **User**

You are given a $896 \times 896$ image that encodes one or more time series variables as horizontal bands stacked from top to bottom. For variable 2, compare cycle 7 and cycle 9 based on overall brightness (0-255 pixel values). These two cycles show visually distinct brightness levels.
Given metadata: $L = 240$, periodicity = 24.
Which cycle has higher values?

Note: Variables are indexed from 1 (top band) to nvars (bottom band).

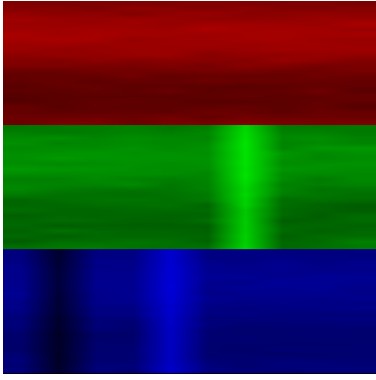

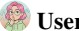 **TIMEOMNI-VL**

`<think>`
**1) Geometry & Layout.**
$n_{\text{vars}} = 3$, band height $h = 298$. Total cycles $N = 10$, cycle width $w = 89.6$. Target Band: **Variable 2** (Middle, Green channel). Y-range: $[298, 595]$.
**2) Region Localization.**
- **Cycle 7:** X-range $[\lfloor 6 \times 89.6 \rfloor, \lfloor 7 \times 89.6 \rfloor - 1] = [537, 626]$.
- **Cycle 9:** X-range $[\lfloor 8 \times 89.6 \rfloor, \lfloor 9 \times 89.6 \rfloor - 1] = [716, 805]$.

**3) Statistical Comparison.**
Calculating mean pixel values (Green channel):
- $\mu_{\text{cyc7}} = 192.8$ (High brightness).
- $\mu_{\text{cyc9}} = 120.9$ (Low brightness).

Difference: $\Delta = |192.8 - 120.9| = 71.9$. Cycle 7 is significantly brighter than Cycle 9.
`</think>`

`Answer:`
7 is brighter (difference: 54.6%).

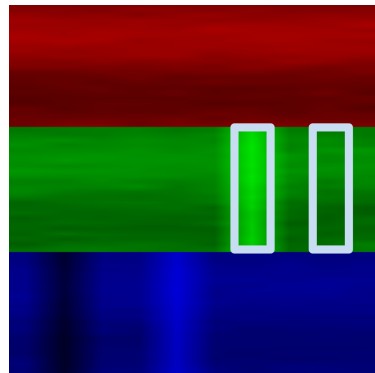

*Table 19.* Example of Understanding Task 5: Anomaly Detection.

---

**Understanding Task 5: Anomaly Detection**

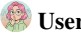 **User**

You are given a $896 \times 896$ image that encodes one or more time series variables as horizontal bands stacked from top to bottom. For variable 3, identify cycles that show significantly abnormal brightness compared to the typical level.

Given metadata: $L = 240$, periodicity = 24.

Definition: "Significantly abnormal" means cycles where the average pixel value (0-255 scale) is either brighter (mean + 18) or darker (mean - 18) than the overall mean for this variable.

Provide the count and bounding boxes (x1, y1, x2, y2) for each anomalous cycle.

Note: Variables are indexed from 1 (top band) to nvars (bottom band).

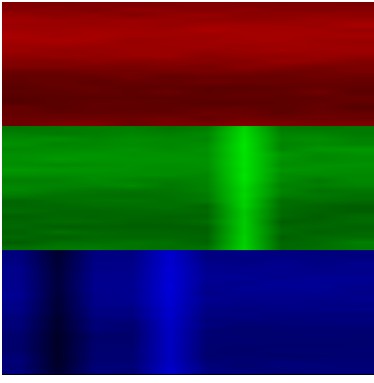

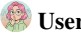 **TIMEOMNI-VL**

`<think>`

**1) Geometry & Layout.**

Target: **Variable 3** (Blue channel). Band height $h = 298$. Y-range $[596, 893]$. Total cycles $N = 10$. Cycle width $w = 89.6$.

**2) Statistical Analysis.**

Computed global mean of 10 cycles: $\mu = 126.6$. Thresholds:

• Bright: $> 126.6 + 18 = 144.6$.

• Dark: $< 126.6 - 18 = 108.6$.

**3) Anomaly Identification.**

Scanning individual cycle averages:

• **Cycle 2:** Mean $67.5 < 108.6 \rightarrow$ Dark Anomaly. Bbox: $[(89, 596), (178, 893)]$.

• **Cycle 5:** Mean $185.7 > 144.6 \rightarrow$ Bright Anomaly. Bbox: $[(358, 596), (447, 893)]$.

`</think>`

`Answer:`

For variable 3, there are 2 anomalous cycles (1 bright, 1 dark). Bounding boxes: $[(89, 596), (178, 893)]$, $[(358, 596), (447, 893)]$.

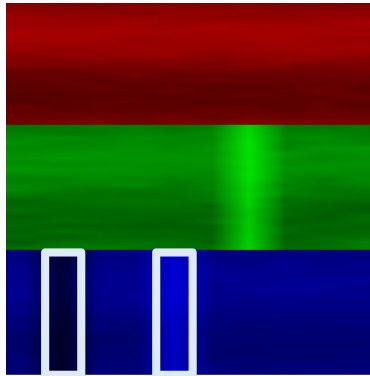

*Table 20.* Example of Understanding Task 6: Trend Analysis.

---

**Understanding Task 6: Trend Analysis**

---

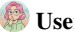 **User**

You are given a $896 \times 896$ image that encodes one or more time series variables as horizontal bands stacked from top to bottom. For variable 3, cycle 10, analyze the time series within this cycle:
1. What color channel is used for this variable?
2. What is the bounding box (x1, y1, x2, y2) of this cycle region?
3. After recovering to original time series values, describe the pattern and trend of this cycle in 2-3 sentences.
Given metadata: $L = 240$, periodicity = 24.
Recovery information:
- Denorm: val $= \tanh(\text{norm}) \cdot 4.0 \cdot \sigma + \mu$
- Mean ($\mu$): 1.52, Std ($\sigma$): 2.77
Note: Variables are indexed from 1 (top band) to nvars (bottom band).

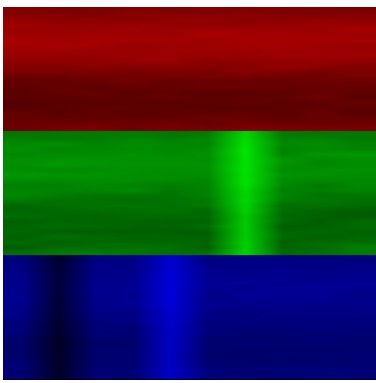

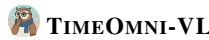 **TIMEOMNI-VL**

<think>
**1) Geometry & Layout.**
Image size $896 \times 896$, $n_{\text{vars}} = 3$. Band height $h = \lfloor 896/3 \rfloor = 298$. Total cycles $N = \lceil 240/24 \rceil = 10$. Cycle width $w = 896/10 = 89.6$ pixels.
**2) Localization.**
Target: **Variable 3** (Bottom), **Cycle 10** (Rightmost).
- Y-range: $[(3 - 1) \times 298, 3 \times 298 - 1] = [596, 893]$.
- X-range: $[\lfloor 9 \times 89.6 \rfloor, \lfloor 10 \times 89.6 \rfloor - 1] = [806, 895]$.
**3) Recovery & Analysis.**
Channel: **Blue** (associated with Var 3). Applying the denormalization formula reveals the raw values. The series starts at 1.57, rises to a peak, and then fluctuates downwards.
</think>

Answer:
1. Color channel: **Blue**.
2. Bounding box: **[(806, 596), (895, 893)]**.
3. Trend analysis: The series initially rises to a prominent peak of 3.22 at $t = 5$, followed by a substantial and fluctuating decline to its minimum of -0.01 at $t = 16$. It then recovers moderately towards the end. Overall, the series shows a slight net decrease.

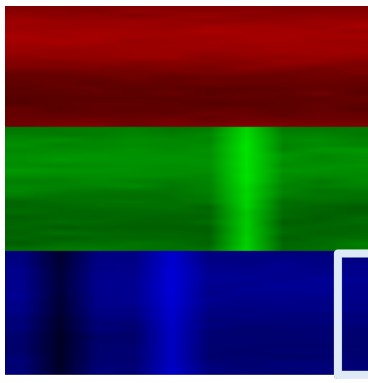

*Table 21.* Example of Generation Task 1: Time Series Forecasting.

**Generation Task 1: Time Series Forecasting**

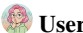 **User**

You are given a $896 \times 896$ image that encodes 2 time series variable(s) as horizontal bands stacked from top to bottom. Each series contains 2 cycles, where each cycle has 288 time steps (totaling $2 \times 288 = 576$ observations). The right side is masked and needs to be predicted for the next 1 cycle (totaling $1 \times 288 = 288$ observations). Based on the observable parts in the time series image, please restore the masked right side.

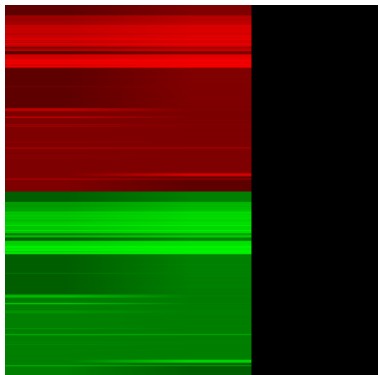

**TIMEOMNI-VL**

```
<think>
```
**1) Variable Counting.**
The image contains 2 independent time series encoded as horizontal bands. Brighter pixels indicate larger values, and darker pixels indicate smaller ones.
**2) Variable Y-Range.**
Total size is $896 \times 896$. With $n_{\text{vars}} = 2$, each band height is $h = 896/2 = 448$ pixels. Total cycles $= 2$ (history) $+ 1$ (pred) $= 3$. Cycle width $\approx 896/3$ pixels.
**3) Cycle Bounding Box.**
The task is to predict the rightmost 3rd cycle.
- **Var 1 (Top):** Pred bbox $\approx [(597, 0), (895, 447)]$.
- **Var 2 (Bottom):** Pred bbox $\approx [(597, 448), (895, 895)]$.

**4) Mean Comparison & Anomaly Detection & Trend Analysis.**
First, analyze the historical pattern (576 steps) for trend and seasonality. Then, extrapolate the next 288 steps for each variable independently. Finally, restore the visual representation maintaining consistent brightness encoding.
```
</think>
```

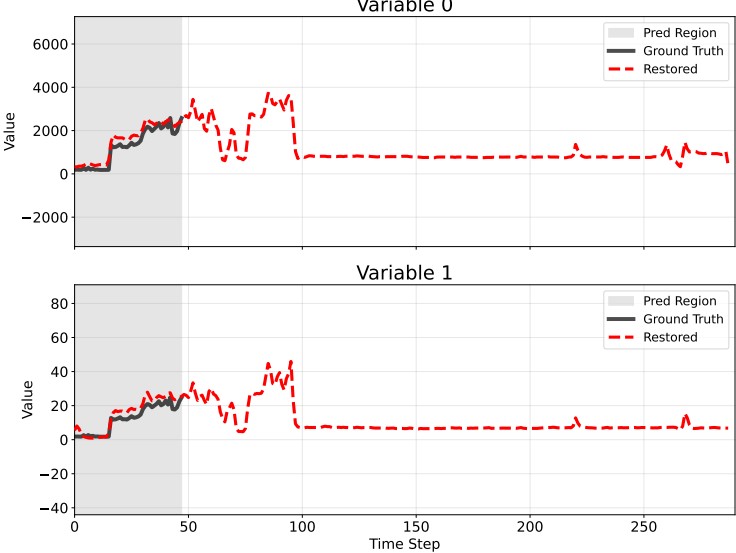

**Special Note:** Following the GIFT-Eval evaluation protocol (Aksu et al., 2024), the required prediction length $(P)$ for this case is 48. Although the model generates a sequence of 288 points, the output is truncated to the first 48 points for quantitative evaluation.

*Table 22.* Example of Generation Task 2: Time Series Imputation.

---

**Generation Task 2: Time Series Imputation**

---

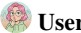 **User**

You are given a 896 × 896 image that encodes 1 time series variable(s) as horizontal bands stacked from top to bottom. Each series contains 6 cycles, where each cycle has 24 time steps (totaling 6 × 24 = 144 observations). Some regions in these series are masked (shown as black areas) and need to be imputed. Different series may have masked regions at different positions, and the number of masked cycles can vary across series. Based on the observable parts in the time series image, please restore all the masked black regions for each series.

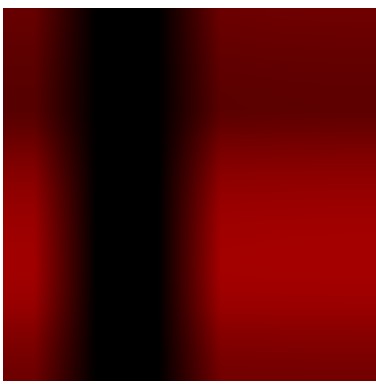

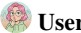 **TIMEOMNI-VL**

`<think>`
**1) Variable Counting.**
The image contains 1 independent time series encoded as horizontal bands. Brighter pixels indicate larger values, and darker pixels indicate smaller ones.
**2) Variable Y-Range.**
Total size is 896 × 896. With $n_{vars} = 1$, the band height is $h = 896$ pixels. Total cycles = 6. Cycle width ≈ 896/6 ≈ 149 pixels per cycle.
**3) Cycle Bounding Box.**
Each series has specific masked black regions.
  • **Var 1:** Missing cycles 2–3. Mask bbox ≈ $[(149, 0), (447, 895)]$.
**4) Mean Comparison & Anomaly Detection & Trend Analysis.**
First, analyze the observable patterns before (cycle 1) and after (cycles 4–6) the gap to identify trend and seasonality. Then, impute the missing cycles (2–3) using the surrounding context, maintaining consistent brightness encoding. Finally, output the restored image.
`</think>`

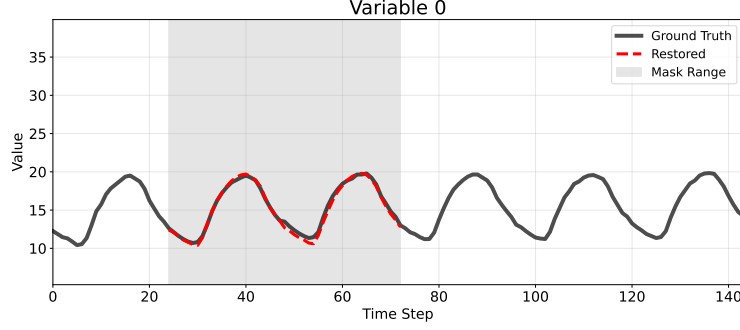

---

### F.2. Comparative Analysis and Failure Cases of the Base Model: Bagel

To further validate the necessity of our time series-specific post-training, we present representative failure cases from our base model, Bagel (Deng et al., 2025), on the same generation tasks. Specifically, Table 23 illustrates a failure in the forecasting task, while Table 24 demonstrates an unsuccessful case for the imputation task.

*Table 23.* Bad Case I: A failure case of Bagel in the time series forecasting task.

---

**Bad Case I: Time Series Forecasting**

---

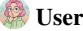 **User**    You are given a 896 × 896 image that encodes 7 time series variable(s) as horizontal bands stacked from top to bottom. Each series contains 4 cycles, where each cycle has 24 time steps (totaling 4 × 24 = 96 observations). The right side is masked and needs to be predicted for the next 2 cycles (totaling 2 × 24 = 48 observations). Based on the observable parts in the time series image, please restore the masked right side.

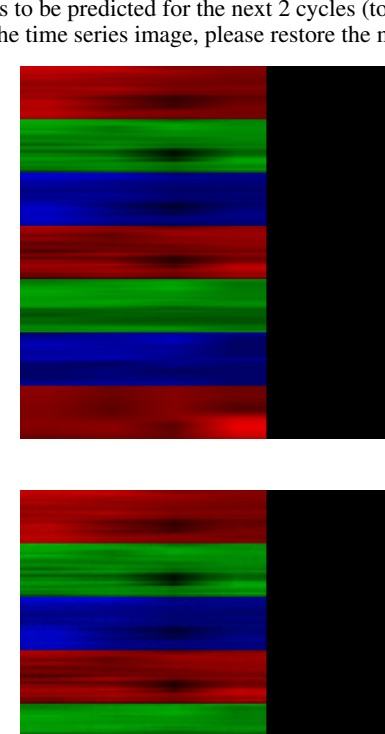

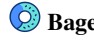 **Bagel**

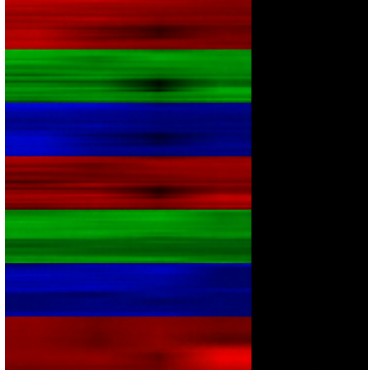

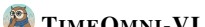 **TIMEOMNI-VL**

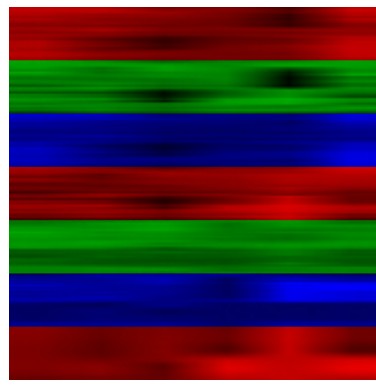

---

*Table 24.* Bad Case II: A failure case of Bagel in the time series imputation task.

---

**Bad Case II: Time Series Imputation**

---

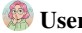 **User**

You are given a 896 × 896 image that encodes 7 time series variable(s) as horizontal bands stacked from top to bottom. Each series contains 24 cycles, where each cycle has 96 time steps (totaling 24 × 96 = 2304 observations). Some regions in these series are masked (shown as black areas) and need to be imputed. Different series may have masked regions at different positions, and the number of masked cycles can vary across series. Based on the observable parts in the time series image, please restore all the masked black regions for each series.

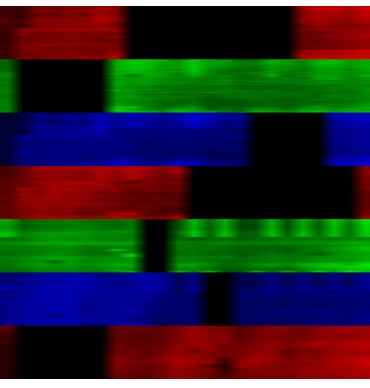

**Bagel**

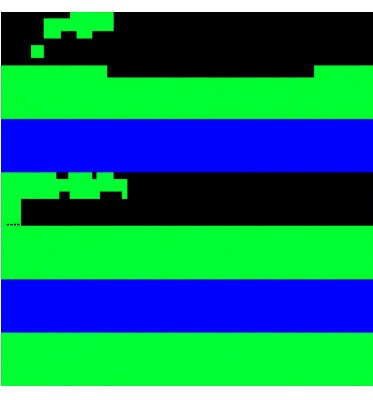

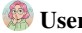 **TIMEOMNI-VL**

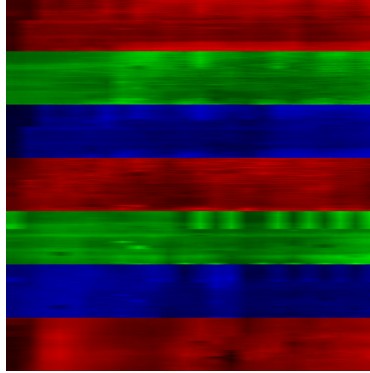

---

## G. Limitations and Future Work

While TIMEOMNI-VL demonstrates the feasibility of vision-centric unified modeling for time series understanding and generation, several limitations remain.

**Task-specific Scaffolding.**  TIMEOMNI-VL is currently built around the proposed TS-image representation. The six TS-image understanding tasks and the derived generation CoT are designed to support forecasting and imputation, and should be viewed as task-grounded scaffolding rather than a universal benchmark for general temporal understanding or a free-form reasoning process. Future work should explore more general forms of temporal reasoning and control signals beyond task-specific CoT construction.

**Information Loss in TS-image Representation.**  Bi-TSI is near-lossless in practice but not strictly lossless: residual errors can still arise from spatial interpolation, finite image resolution, and numerical precision. Moreover, the current design relies on capacity constraints, multi-color band rendering, and relatively high-resolution TS-images, which introduce an efficiency trade-off and may limit direct applicability to single-color or irregularly sampled time series without additional adaptation. Future iterations should develop more efficient and adaptive TS-image encodings while preserving numerical fidelity.

**Scope and Scalability.**  Our experiments instantiate the framework with Bagel as the backbone and use limited generation training data; thus the results should be interpreted as establishing a unified modeling paradigm with competitive performance rather than uniformly outperforming specialized time series models at every horizon. Future work will explore more UMM backbones, broader external benchmarks, and native support for irregular time series.

