# OpenReview forum: "TimeOmni-VL: Unified Models for Time Series Understanding and Generation"
_ICML.cc/2026/Conference — ICML 2026 regular_

### Official Review · Reviewer_5jTa · 2026-03-02

**Soundness:** 3
**Presentation:** 3
**Significance:** 3
**Originality:** 3
**Overall Recommendation:** 5
**Confidence:** 4

**Summary:**

Aiming at dealing with the gap between numerical value generation and semantic understanding in current time series modeling, this paper proposes the first vision-centric unified framework, TIMEOMNIVL. This framework enables high-fidelity numerical output through a nearly lossless fidelity bidirectional mapping between time series and images (Bi-TSI), and comprehension guided generation based on a newly constructed dataset with CoT - which for the first time uses semantic understanding as an explicit control signal. Experiments show that this method effectively exploits the potential of multimodal models in the field of time series, improves the semantic understanding ability and numerical accuracy of modeling, and it provides a new idea for the understanding-generating unified time series modeling.

**Compliance With Llm Reviewing Policy:**

Affirmed.

**Final Justification:**

This paper has novel ideas and solid experiments.  After rebuttal, it solves all my problems and gives insight into the future direction, so I choose to raise my score to accept

**Key Questions For Authors:**

Q1: Compared with VisionTS++, this paper uses a higher image resolution and a larger model, but does not achieve better results. Is the improvement brought by sacrificing efficiency worth it?

Q2: What is the root cause of the inferior medium and long term forecast to VisionTS++? Please discuss in detail.

Q3: The VisionTS family of models has powerful zero-shot and few-shot predictive power. Explore or discuss: Does this method have similar power?

**Limitations:**

See Weaknesses.

**Strengths And Weaknesses:**

Strength
- Novelty. Prior to this method, the mainstream practice of temporal unified modeling was to wire shallow MLPS on LLMS and forecast with directly flattened or weighted text tokens. This paper finds a more natural common way. In fact, with the help of VLMs, the understanding representation of the time-series data image can be directly used as the guidance and condition for generation, and understanding (image analysis) and generation (image inpainting) naturally coexist in the same representation space, without cross-modal alignment. The method is closed-loop and intuitive.
- The dataset is novel in design and valuable for the community to carry out similar unified research on understanding generation.
- The clarity is good.

Weaknesses
- The medium and long-term prediction is worse than VisionTS++. Among the 4 reasoning tasks, only Task4 is the best.
- Lack of detailed efficiency analysis compared to other methods.
- Performance still has improvement room.

---

> ### Author Rebuttal · Authors · 2026-03-31
>
> > **W1 & W3 & Q2:** Performance has improvement room.
>
> Thanks for the insightful comments. We would like to clarify three aspects.
>
> **(1) Research Objective: Unified Framework vs. Specialized SOTA**
>
> Our objective is not to outperform every specialized forecaster at every horizon, but to establish a **unified framework** that effectively integrates time series understanding and generation.
>
> More importantly, the competitive results already validate the feasibility of a new paradigm for time series, namely **understanding-guided generation**, even with marginal performance trade-offs.
>
> **(2) Forecasting Performance (Med/Long-term)**
>
> - **Limited training data.** The current results use only 5k training samples per generation task. To isolate this effect, we retrained VisionTS++ on the **same 5k samples**. Results show that TimeOmni-VL outperforms VisionTS++, indicating competitiveness under equal data conditions.
>
>   Detailed results: https://anonymous.4open.science/r/19113/T2.png. For convenience, we summarize below:
>
> |*Metric: nMASE*|Fore. AVG (GIFT-Eval)|Impu. AVG (GIFT-Eval)|
> |-|-|-|
> |VisionTS++ (retrained on our training set)|0.900|0.827|
> |TimeOmni-VL|**0.826**|**0.810**|
>
> - **Benchmark-specific variation.** We additionally evaluate on the third-party **TIME** [1] benchmark, where TimeOmni-VL is stronger in medium-term forecasting. This suggests that the original inferiority is at least **partly affected by benchmark characteristics**.
>
> |*Metric: nMASE*|Fore. Short (TIME)|Fore. Med (TIME)|Fore. Long (TIME)|
> |-|-|-|-|
> |VisionTS++ (Original ckpt)|0.830|0.921|**0.836**|
> |VisionTS++ (retrained on our training set)|0.833|0.926|1.008|
> |TimeOmni-VL|**0.815**|**0.832**|0.926|
>
> **(3) Reasoning Performance**
> Current results reported in the paper rely solely on Stage-1 Supervised Fine-Tuning data from TSR-Suite to inject temporal priors. We did not utilize the Stage-2 Reinforcement Learning data (16k samples), which typically boosts complex reasoning. This leaves clear room for improvement.
>
> [1] *It's TIME: Towards the Next Generation of Time Series Forecasting Benchmarks*, 2026
>
> ---
> > **W2:** Lack of efficiency analysis.
>
> Thanks for pointing this out.
>
> - The efficiency of TimeOmni-VL is primarily determined by the UMM backbone (Bagel), rather than our pipeline alone. Meanwhile, Bi-TSI only involves simple tensor operations with negligible overhead.
> - A direct efficiency comparison across methods is difficult, because runtime depends heavily on implementation details and hardware, especially when baselines (e.g., Gemini-2.5) are proprietary and do not expose comparable inference budget.
> - To provide a controlled view of efficiency, we report the **performance-efficiency trade-off under different TS-image resolutions**. See response to **Q1**.
>
> ---
> > **Q1:**  Why higher resolution and larger model?
>
> Thanks for raising this question.
>
> **(1) Higher Resolution Serves Encoding Capacity Control.**
>
> The choice of an $896×896$ resolution is a deliberate design for **Encoding Capacity Control** in $\underline{\text{Section 3.1}}$. This provides 16× more area than VisionTS++’s $224×224$ resolution, allowing TimeOmni-VL to encode more variables and longer sequences without introducing downsampling.
>
> To fully address your concern, we evaluate performance and efficiency across three TS-image resolutions. Detailed results: https://anonymous.4open.science/r/19113/T9.png. We summarize below:
>
> |Resolution|Train / Image|Inference / Image|Fore. AVG|Impu. AVG|
> |-|-|-|-|-|
> |**224 × 224**|44 ms|11s|0.851|0.874|
> |**448 × 448**|149 ms|24s|0.810|0.838|
> |**896 × 896**|651 ms|50s|**0.783**|**0.798**|
>
> **Findings:**
>
> - **Higher Resolution → Higher performance.**
>
> - **Performance–Efficiency Trade-off.** $896×896$ balances information preservation and computational cost.
>
> **(2) Larger Model.**
>
> TimeOmni-VL is built on **unified multimodal models (UMMs)**. Among existing UMMs (e.g., EMU2 34B, BLIP3‑o 8B, TokenFlow‑XL 14B), **Bagel‑7B is relatively lightweight within our training budget.**
>
> Our broader goal is to **internalize** time series understanding and generation as **native capabilities of UMMs**, enabling time series data to become part of UMMs' training corpora rather than developing another time series specific model. We view this as a necessary step forward, even with moderate efficiency costs.
>
> ---
> > **Q3:**  have zero-shot power?
>
> We would like to clarify that all results in the paper are obtained **under a strict zero-shot evaluation**; the model never sees the GIFT-Eval testbed during training.
>
> To further address your concern, we additionally evaluate TimeOmni-VL on the third-party **TIME** [1] benchmark. Detailed results: https://anonymous.4open.science/r/19113/T5&6.png. We summarize below:
>
> |*Metric: nMASE*|Fore. AVG (TIME)|Impu. AVG (TIME)|
> |-|-|-|
> |VisionTS++|0.862|1.036|
> |TimeOmni-VL|**0.858**|**0.784**|
>
> **Finding.** These results confirm that TimeOmni-VL remains effective beyond GIFT-Eval.
>
> [1] *It's TIME...*, 2026.

---

> > ### Author Rebuttal · Reviewer_5jTa · 2026-04-02
> >
> > Thanks for your rebuttal, my concerns have been fully solved, so I choose to rasie my score to accept.

---

> > > ### Author Response · Authors · 2026-04-02
> > >
> > > We are deeply grateful for your prompt response and your positive assessment, which are extremely important and encouraging to us.
> > >
> > > Best regards,
> > >
> > > The Authors

---

### Official Review · Reviewer_6ZUV · 2026-03-07

**Soundness:** 4
**Presentation:** 4
**Significance:** 3
**Originality:** 3
**Overall Recommendation:** 5
**Confidence:** 4

**Summary:**

This paper introduces a unified large model for time series, meaning a single model that can handle both time series understanding and generation tasks. The paper presents a new benchmark, TSUMM-SUITE, and a new model, TIMEOMNI-VL. By jointly training on both types of tasks and leveraging understanding-CoT-guided generation, the model bridges two types of tasks that are traditionally treated separately. Experimental results show that the proposed model achieves good performance on the introduced time series benchmarks, particularly on understanding tasks.

**Compliance With Llm Reviewing Policy:**

Affirmed.

**Final Justification:**

I have no further questions and I score 5 as the final score of this paper.

**Key Questions For Authors:**

Though the overall quality of this paper is relative high, I still have questions below:
1. Could the authors provide results with different backbone models to demonstrate how sensitive performance is to the backbone choice and to better support the generalization claim?
2. How robust is the model if the order of multiple time series in a single image is shuffled?
3. If the same color is used to render all series, would the understanding and generation tasks fail?
4. What is the impact of different TS-image sizes?
5. Is it possible to evaluate the model on third-party time series tasks, such as MTBench (MTBench: A Multimodal Time Series Benchmark for Temporal
   Reasoning and Question Answering) benchmark adapted for time series?
6. Although the model covers a much broader range of tasks than existing TSFM, it would be helpful to better understand how large the performance gap is compared to current TSFM.

**Limitations:**

Adding a section to discuss the limitations would better strengthen the paper.

**Strengths And Weaknesses:**

### Strengths

1. The paper clearly identifies the current polarization in time series analysis between understanding models and generation models, and on this basis proposes a unified model, namely TIMEOMNI-VL. I think unified models are needs in current time series community.
2. The analysis of existing TS2I strategies is thorough and well illustrated (Fig 3). The appendix D provides additional details on key comparisons, which further improve its clarity.
3. Although vision-based approaches for time series analysis are not entirely new, the use of understanding-CoT-guided generation to combine two tasks that are typically treated as separate is innovative. The experimental results show that the proposed CoT mechanism leads to noticeable performance gains on both generation tasks.
4. The paper designs six understanding tasks and two generation tasks, going beyond the task coverage of most existing time series models and providing an exploratory step toward unified time series modeling. The comparison of different TS2I strategies is informative and helps justify the design choice. The ablation studies are appropriate.

### Weaknesses

1. Although the paper claims that the framework is backbone-agnostic, it should report the impact of different backbones (e.g. Janus models) on performance in order to substantiate its generalization.
2. The proposed TS2I method builds upon VisionTS-Series, but some limitations on VisionTS-Series still seem to persist and are not fully discussed. For example, how robust is the model if the order of multiple time series in a single image is shuffled? If the same color is used to render all series, would the understanding and generation tasks fail?
3. The paper appears to use images of size 896 × 896. In common practice, this is already a relatively large resolution, yet the potential impact of TS-image size on performance and efficiency is not discussed.
4. The paper should include a section that discusses the limitations.

---

> ### Author Rebuttal · Authors · 2026-03-31
>
> > **W1 & Q1:** Could the authors provide results with different backbone models to demonstrate how sensitive performance is to the backbone choice?
>
> Thank you for this insightful comment. Adding another backbone is training-intensive and not feasible within the rebuttal timeline, but we will include an additional backbone in the revision if feasible.
>
> ---
>
>
>
> > **W2-part1 & Q2:** How robust is the model if the order of multiple time series in a single image is shuffled?
>
> Thanks for pointing this out. To speed up evaluation, we randomly sample 200 multivariate test samples from the GIFT-Eval testbed for each task (forecasting and imputation). We then **shuffle the order of the multiple time series within each TS-image to create three distinct evaluation sets** (Shuffle Groups 1–3). Detailed results are provided in https://anonymous.4open.science/r/19113/T7.png. For convenience, we report the 95% confidence interval (CI) below:
>
> |Forecasting|Mean ± 95% CI|
> |-|-|
> |Short|0.811 ± 0.018|
> |Med|0.794 ± 0.035|
> |Long|0.805 ± 0.042|
>
> |Imputation|Mean ± 95% CI|
> |-|-|
> |[0.1,0.2)|0.631 ± 0.075|
> |[0.2,0.3)|0.890 ± 0.102|
> |[0.3,0.4)|0.922 ± 0.055|
> |[0.4,0.5]|0.948 ± 0.058|
>
> **Finding.** The model remains highly robust to random variable ordering.
>
> ---
>
>
>
> > **W2-part2 & Q3:** If the same color is used to render all series, would the understanding and generation tasks fail?
>
> Many thanks for this careful observation. To speed up evaluation, we use the same test samples as in our response to **W2-part1**. We then **render all time series in a single color** and evaluate under two settings:
>
> - **Inference-only:** directly test the original TimeOmni-VL on single-color images.
> - **Fine-tuning:** retrain the base model (Bagel-7B) using single-color images.
>
> Detailed results are provided in https://anonymous.4open.science/r/19113/T8.png. For convenience, we summarize below, where **Bold** means best.
>
> |*Metric: nMASE*|Fore. AVG|Impu. AVG|Und. AVG|
> |-|-|-|-|
> |Inference-only (single color)|0.955|0.886|0.707|
> |Fine-tuning (single color)|0.816|0.824|0.893|
> |Ours (multi-color)|**0.783**|**0.798**|**0.908**|
>
> **Findings.**
>
> **(1) Current TimeOmni-VL relies on the multi‑color setting.** Direct inference on single‑color images leads to performance drops, likely because the model struggles to distinguish band boundaries without color cues.
>
> **(2) Multi-color rendering is still more effective.** Even after fine-tuning on single-color images, the performance remains below the original multi-color setting.
>
> ---
>
>
>
> > **W3 & Q4:** What is the impact of different TS-image sizes?
>
> Thanks for pointing this out. As we clarified in $\underline{\text{Section 3.1}}$, the larger $896×896$ resolution is used to encode more variables and longer sequences without introducing downsampling.
>
> To fully address your concern, we evaluate both performance and efficiency under three TS-image resolutions. Detailed results are provided in https://anonymous.4open.science/r/19113/T9.png. For convenience, we summarize the efficiency metrics and nMASE results below.
>
> |Resolution|Train / Image|Inference / Image|Fore. AVG|Impu. AVG|
> |-|-|-|-|-|
> |**224 × 224**|44 ms|11s|0.851|0.874|
> |**448 × 448**|149 ms|24s|0.810|0.838|
> |**896 × 896**|651 ms|50s|**0.783**|**0.798**|
>
> **Findings:**
>
> **(1) Higher Resolution → Higher performance.** Both forecasting and imputation improve steadily as the TS-image size increases from $224×224$ to $896×896$.
>
> **(2) Performance–Efficiency Trade-off.** Larger resolutions require substantially more training and inference time, $896×896$ reflects our trade-off between information preservation and computational cost.
>
> ---
>
>
>
> > **W4 & Limitations:** Adding a section to discuss the limitations would better strengthen the paper.
>
> We sincerely appreciate for raising this important point. We will **add a dedicated Limitation section** in the next revision.
>
> ---
>
>
>
>
> > **Q5:** Is it possible to evaluate the model on third-party time series tasks?
>
> Due to the strict character limit of the ICML rebuttal, we kindly refer you to the following additional results:
>
> - **Third-party understanding benchmark (MTBench): Response to W3 of Reviewer LcZS.**
> - **Third-party generation benchmark (TIME): Response to Q3 of Reviewer 5jTa.**
>
> ---
>
>
>
>
> > **Q6:** Although the model covers a much broader range of tasks than existing TSFM, it would be helpful to better understand how large the performance gap is compared to current TSFM.
>
> Thanks for pointing this out. Please refer to $\underline{\text{Table 1}}$ and $\underline{\text{Table 2}}$ for the comparison with TSFMs (VisionTS and Moment Series). Our goal is to establish a unified framework that covers both time series understanding and generation. Existing TSFMs are typically optimized for a much narrower scope, whereas **TimeOmni-VL aims to support a broader range of tasks within a single model while maintaining competitive performance.**

---

> > ### Author Rebuttal · Reviewer_6ZUV · 2026-04-01
> >
> > Thanx for the response and I have no further questions.

---

> > > ### Author Response · Authors · 2026-04-02
> > >
> > > We are very glad to hear that our rebuttal has addressed your concerns. Thank you again for the detailed and constructive comments, which have helped us further strengthen this work.
> > >
> > > We are also deeply grateful for your strong support and positive evaluation, which are extremely important and encouraging to us.
> > >
> > > Best regards,
> > >
> > > The Authors

---

### Official Review · Reviewer_tuqx · 2026-03-11

**Soundness:** 3
**Presentation:** 3
**Significance:** 2
**Originality:** 3
**Overall Recommendation:** 4
**Confidence:** 4

**Summary:**

This paper studies whether time-series understanding and generation can be handled within a single multimodal framework. The authors propose TIMEOMNI-VL, a vision-based approach built on a bidirectional time-series/image representation pipeline (Bi-TSI) together with an understanding-guided generation scheme. In the proposed design, generation is guided by a chain-of-thought-style signal ($R_{gen}$) derived from instance-level understanding tasks. The paper also introduces TSUMM-SUITE, a benchmark suite consisting of two generation tasks, forecasting and imputation, along with six understanding tasks defined on top of the proposed time-series image representation.

**Compliance With Llm Reviewing Policy:**

Affirmed.

**Final Justification:**

Updated my score to weak accept after author rebuttals that addressed all my concerns.

**Key Questions For Authors:**

The main issues raised above in the weaknesses section are potentially addressable, and the paper would be significantly strengthened if the authors could respond to them more directly. The paper would benefit from a clearer disentangling of the TS2I design and the proposed $R_{gen}$ mechanism. Stronger controls, more complete baseline coverage, and a fuller discussion of validity, robustness, and scalability would also improve the paper. I highly  encourage the authors to address these issues in the rebuttal, since doing so would improve the overall quality and credibility of the paper. If the rebuttal convincingly addresses these concerns, I would be willing to raise my score.

**Limitations:**

No. The paper discusses some technical constraints of the TS2I design, but I believe the limitations should be stated more explicitly in the main paper.

**Strengths And Weaknesses:**

Stengths:
1. The paper tackles a meaningful problem: whether time-series understanding and generation can be handled within a single multimodal model.
2. The Bi-TSI design is more carefully thought out than a straightforward heatmap-style conversion. The treatment of robust fidelity normalization and the periodicity-aware TS2I/I2TS mapping feels like a nontrivial part of the technical contribution.
3. The experimental results on the understanding tasks are quite strong, especially compared with the untuned backbone. The imputation results are also impressive and appear consistently solid across different masking ranges.
4. I also think the idea of using understanding-derived semantic guidance to support generation is interesting. Even though I have some reservations about how convincingly the mechanism is validated, the direction itself is promising and worth exploring further.

Weaknesses:
1. My main concern is that several key ingredients are introduced at the same time: the proposed TS2I representation, a task suite designed around that representation, and the generation CoT derived from understanding tasks constructed from the same generation instances. Because these pieces are so tightly coupled, it is difficult to tell how much of the reported gain really comes from the semantic role of $R_{gen}$, as opposed to the overall design package.
2. Relatedly, I do not think the current ablations fully establish that $R_{gen}$ helps because of its semantic content. At the moment, it is still possible that part of the gain comes simply from adding more context, more structure, or another form of supervision. Stronger controlled experiments would make this aspect much more convincing.
3. The forecasting results are good, but they are not uniformly the best among the compared methods. TIMEOMNI-VL performs slightly better than VisionTS++ in the short-term setting, while VisionTS++ remains stronger in the medium- and long-term settings. Because of this, I would describe the forecasting evidence as competitive rather than clearly superior.
4. I also found the baseline coverage somewhat uneven across tasks. VisionTS++ is included in the forecasting comparison, but not in the imputation table. Even if there is a reasonable explanation for this, the current presentation leaves the empirical comparison feeling incomplete.
5. Another issue is that the reported metrics are based on valid and extractable outputs, and results are omitted when the success rate falls below a threshold. This makes the main numbers harder to interpret from a practical perspective, since output validity and reliability are themselves important parts of the model’s usefulness.
6. The evidence for generalization on the generation side also feels somewhat narrow. Since the training and evaluation pipeline is centered on the GIFT-Eval family, it is hard to know how well the approach transfers beyond that benchmark setting.
7. Finally, the TS2I construction seems to work under a relatively favorable regime where dimensionality and periodic structure remain manageable with respect to image resolution. That seems fine for the current experiments, but I am less sure how well the approach would scale to substantially higher-dimensional or less regular time series.

---

> ### Author Rebuttal · Authors · 2026-03-31
>
> > **W1 & W2:** Stronger controlled experiments.
>
> Thanks for your insightful comments. To directly test whether the gain of $R_{\text{gen}}$ comes from its semantic content, we introduce three controlled variants:
>
> - **Context Mismatch**: replace $R_{\text{gen}}$ with a CoT from a different training sample, **keeping context but mismatching semantics**;
> - **Structural Template Only**: use a generic template CoT with no instance-specific content, **keeping structure but ablating semantic details**;
> - **Task-Irrelevant Supervision**: supervise the model with a CoT irrelevant to the target task, **keeping supervision pathway but breaking task alignment**.
>
> Detailed results: https://anonymous.4open.science/r/19113/T1.png. Summary below:
>
> |*Metric: nMASE*|Fore. AVG|Impu. AVG|
> |:--|-|-|
> |Context Mismatch|0.921|0.933|
> |Structural Template Only|0.870|0.846|
> |Task-Irrelevant Supervision|1.059|0.949|
> |Original $R_{\text{gen}}$|**0.826**|**0.810**|
>
> **Findings.** All three variants underperform the original $R_{\text{gen}}$. This confirms that the gain requires the correct instance- and task-grounded semantics in $R_{\text{gen}}$.
>
> ---
> > **W3:** describe the forecasting evidence as competitive.
>
> We **will revise the wording accordingly**. Our goal is not to outperform every specialized forecaster at every horizon, $\underline{\text{but to establish a unified framework for both time series understanding and generation with strong performance}}$. We also note that the current results use only **5k training samples per generation task**. To isolate this effect, we retrained VisionTS++ **on the same 5k samples**. Detailed results: https://anonymous.4open.science/r/19113/T2.png. Summary below:
>
> |*Metric: nMASE*|Fore. AVG|Impu. AVG|
> |-|-|-|
> |VisionTS++ (retrained on our training set)|0.900|0.827|
> |TimeOmni-VL|**0.826**|**0.810**|
>
> **Findings.** Under equal data conditions, TimeOmni-VL consistently outperforms VisionTS++, supporting our understanding-guided generation paradigm.
>
> ---
> > **W4:** baseline coverage.
>
> We have now **adapted VisionTS++ to the imputation task** and retrained it accordingly. To ensure a fair comparison, we use the same training data and epochs. Detailed results: https://anonymous.4open.science/r/19113/T2.png. Summary below:
>
> |*Metric: nMASE*|Impu. AVG|
> |-|-|
> |VisionTS++ (retrained for imputation)|0.827|
> |TimeOmni-VL|**0.810**|
>
> **Finding.** Under this controlled comparison, TimeOmni-VL outperforms VisionTS++ on imputation, mainly because explicit semantic guidance improves data efficiency.
>
> ---
> > **W5:** output validity.
>
> Due to space limits, success rates (SR) were omitted in the initial submission. We now provide full results in https://anonymous.4open.science/r/19113/T3&4.png. SR for three representative models are below:
>
> ||Fore. SR %|Impu. SR %|
> |-|-|-|
> |Gemini-2.5-flash|79.3|85.7|
> |Qwen2.5-Instruct-7B|51.1|72.2|
> |TimeOmni-VL|**84.7**|**89.9**|
>
> ---
> > **W6:** The evidence for generalization.
>
> We would like to clarify that all results in the paper are obtained **under a strict zero-shot evaluation**; the model never sees the GIFT-Eval testbed during training.
> To further address your concern, we additionally evaluate TimeOmni-VL on the **TIME** [1] benchmark. Detailed results: https://anonymous.4open.science/r/19113/T5&6.png. Summary below:
>
> |*Metric: nMASE*|Fore. AVG (TIME)|Impu. AVG (TIME)|
> |-|-|-|
> |VisionTS++|0.862|1.036|
> |TimeOmni-VL|**0.858**|**0.784**|
>
> **Finding.** TimeOmni-VL remains effective beyond GIFT-Eval.
>
> [1] *It's TIME: ...Time Series Forecasting Benchmarks*, 2026
>
> ---
> > **W7:** scale to higher-dimensional or less regular time series.
>
> We add two stress tests:
>
> **(1) Higher-dimension (HD).** We construct **10/20/30-dimensional** testbeds from different source datasets, so only same-dimension comparisons are meaningful.
>
> |*Metric: nMASE*|10-D|20-D|30-D|
> |-|-|-|-|
> |VisionTS++|0.866|0.773|0.960|
> |TimeOmni-VL|**0.769**|**0.687**|**0.840**|
>
> **Finding.** TimeOmni-VL consistently performs stronger, mainly because our **Encoding Capacity Control** design avoids downsampling in HD scenarios.
>
> **(2) Less regular (LR).** For clarity, we interpret this as irregular time series (IrTS). **We also acknowledge that native handling of IrTS is out of scope.**
>
> We first interpolate IrTS [1] onto a regular grid, then build a forecasting task for comparison.
>
> |*Metric: nMASE*|LR Fore.|
> |-|-|
> |VisionTS++|1.15|
> |TimeOmni-VL|1.38|
>
> **Finding.** Both models show **nMASE > 1**, indicating that irregular forecasting remains challenging.
>
> [1] *Predicting in-hospital mortality of icu patients*, 2012
>
> ---
> > **Q & Limitations:**.
>
> **(1) TS2I** is completed **before the data enters the backbone model**. In contrast, $R_{\text{gen}}$ operates **within the backbone** as a semantic control signal for generation.
>
> **(2) We respond to the weaknesses as follows:**
>
> - Stronger controls→W1/W2
> - Baseline coverage→W4
> - Validity→W5
> - Robustness→W6
> - Scalability→W7
>
> **(3) We will add a Limitation section** in the revision.

---

> > ### Author Rebuttal · Reviewer_tuqx · 2026-04-02
> >
> > I thank the authors for their thorough rebuttal and for providing the additional experimental results. Based on these clarifications and the strong empirical evidence provided in the rebuttal, my concerns are fully resolved. I have adjusted my score accordingly to reflect the paper’s contribution to unified time series modeling.

---

> > > ### Author Response · Authors · 2026-04-02
> > >
> > > We are very glad to hear that our rebuttal has addressed your concerns. Thank you again for the detailed and constructive comments, which have helped us further strengthen this work.
> > >
> > > We are also deeply grateful for your strong support and positive evaluation, which are extremely important and encouraging to us.
> > >
> > > Best regards,
> > >
> > > The Authors

---

### Official Review · Reviewer_LcZS · 2026-03-15

**Soundness:** 3
**Presentation:** 2
**Significance:** 2
**Originality:** 3
**Overall Recommendation:** 5
**Confidence:** 3

**Summary:**

The paper proposes VLM based framework for time series understanding and generation. The key ideas is a bidirectional mapping between time series and images, and the unified understanding model which can be used for time series QA or understanding guided generation.

**Compliance With Llm Reviewing Policy:**

Affirmed.

**Final Justification:**

The rebuttal has addressed my concerns on the paper. and I choose to raise my score to accept.

**Key Questions For Authors:**

See above weakness

**Limitations:**

yes

**Strengths And Weaknesses:**

Strengths:
1. The idea of utilizing UMMs for time series by converting them to images sounds reasonable.
2. The time series - image conversion pipeline is more carefully designed than a naive plotting-based approach, and the bidirectional mapping is a meaningful engineering contribution.
3. The unified framework is widely applicable to various time series tasks including understanding, forecasting, imputation, and reasoning.

Weaknesses:
1. The practical value of converting time series into large images and then applying a UMM is not fully convincing. While the setup is technically interesting, the paper does not clearly establish when this is preferable to native time-series models beyond making multimodal modeling possible.
2. The bidirectional mapping is described as near-lossless rather than lossless, and the paper acknowledges residual reconstruction error. As a result, the fidelity claim should be interpreted more cautiously.
3. Most of the tasks appear tightly coupled to the authors’ own image representation, so it is unclear how much they measure general temporal understanding rather than proficiency on the custom rendering scheme.
4. The generation CoT is also not independently grounded, which is directly derived from tasks coupled to the proposed TS-image representation and appears to be more benchmark-specific scaffolding rather than a natural reasoning process

---

> ### Author Rebuttal · Authors · 2026-03-31
>
> > **W1:** While the setup is technically interesting, the paper does not clearly establish when this is preferable to native time-series models beyond making multimodal modeling possible.
>
> Many thanks for raising this question. **Multimodal modeling is preferable as follows:**
>
> **(1) Broader input modality and task scope.** Native time series models are mainly designed for numerical signal alone, whereas TimeOmni-VL naturally supports **time series together with auxiliary modalities such as text signal**. This is preferable **when** the downstream task depends not only on the raw series itself, but also on external semantic information [1].
>
> **(2) Unlocking multimodal understanding for controllable generation.** Native time series models often lack explicit semantic understanding. By converting time series into TS-images, we unlock unified multimodal models' (UMMs) **visual understanding capability as an explicit control signal for guiding generation**. This view is also **supported by Reviewer 5jTa**:
>
> >> **Reviewer 5jTa:** "Novelty. Prior to this method, the mainstream practice of temporal unified modeling was to wire shallow MLPS on LLMS and forecast with directly flattened or weighted text tokens. **This paper finds a more natural common way**... understanding (image analysis) and generation (image inpainting) **naturally coexist in the same representation space**, without cross-modal alignment. **The method is closed-loop and intuitive.**"
>
> **(3) Better scaling potential for long-context modeling.** For native time series models, the number of input tokens/patches grows with the number of timesteps, so the backbone input scales with context length. In contrast, in our multimodal modeling, the number of visual tokens depends only on image size and remains fixed. **This allows longer time series inputs to be encoded into the same backbone token budget**, offering better scaling potential for long-context modeling.
>
> [1] *Context is key: A benchmark for forecasting with essential textual information*, 2024
>
> ---
>
>
>
> > **W2:** the fidelity claim should be interpreted more cautiously.
>
> Many thanks for this careful observation. We agree that **near-lossless should not be equated with perfect fidelity**. Our use of the term ***fidelity-preserving*** was only intended to indicate the improvement over existing VisionTS++'s converter, rather than to claim a strictly lossless mapping. **We will make this distinction between *fidelity-preserving* and *strictly lossless* more explicit in the revision.**
>
> ---
>
>
>
> > **W3:** how much they measure general temporal understanding rather than proficiency on the custom rendering scheme.
>
> Many thanks for raising this question.
>
> **(1) Design intent of TS-image tasks.**
>
> Our six TS-image understanding tasks are **not meant to be a universal benchmark for general temporal understanding**, they are deliberately constructed to derive $R_{\text{gen}}$ and provide useful CoT guidance for downstream generation.
>
> **(2) The custom TS-image is just an internal data-augmentation mechanism inside TimeOmni-VL.**
>
> From the overall model perspective, the system still **takes time series as input and produces task outputs as the final objective**. Crucially, **general temporal understanding is decoupled from TS-image**: temporal priors are injected via pure-text TSR-Suite dataset during training, and evaluated via pure-text inference.
>
> **(3) We further evaluate TimeOmni-VL on an external temporal understanding benchmark** (**MTBench** [1]).
>
> In this setting, inference is conducted in a **pure-text format**, rather than through TS-images. Results are listed below (**Bold**: the best),
>
> ||FinanceQA (7-Day)|WeatherQA (7-Day)|FinanceQA (30-Day)|WeatherQA (14-Day)|AVG|
> |-|-|-|-|-|-|
> |*Num of Samples*|*484*|*666*|*523*|*707*|*—*|
> |**Original Paper Result** (GPT-4o)|65.1|41.7|52.8|**44.8**|51.1|
> |**TimeOmni-VL**|**70.2**|**52.0**|**62.1**|44.0|**57.1**|
>
> **Finding.** TimeOmni-VL’s strong performance on MTBench demonstrates that its learned time series priors generalize across domains and tasks.
>
> [1] *MTBench: A Multimodal Time Series Benchmark for Temporal Reasoning and Question Answering*, 2025.
>
> ---
>
>
>
> > **W4:** The generation CoT is ... benchmark-specific scaffolding rather than a natural reasoning process.
>
> We sincerely appreciate this insightful point. We agree that the generation CoT $R_{\text{gen}}$ is better viewed as **task-specific scaffolding**, rather than a universally natural reasoning process. This is exactly our design goal: $R_{\text{gen}}$ is introduced not to perform general free-form reasoning, but to **provide an explicit, task-grounded control signal for downstream generation tasks** (forecasting/imputation). Our goal is to verify whether **understanding-guided generation** can work effectively in time series domain.
>
> Additionally, **we will add a limitation section in the revision to clarify this task-specific scaffolding more explicitly.**

---

> > ### Author Rebuttal · Reviewer_LcZS · 2026-04-02
> >
> > Thanks the authors for their response. My concerns on the paper have been addressed or acknowledge by the rebuttal, and I have no further questions.

---

> > > ### Author Response · Authors · 2026-04-02
> > >
> > > We are truly delighted that our responses have addressed your concerns. We would like to once again express our sincere gratitude for taking the time to review our paper and for providing such insightful and invaluable comments.
> > >
> > > Best regards,
> > >
> > > The Authors

---

### Decision · Program_Chairs · 2026-04-30

**Decision:**

Accept (regular)

**Comment:**

This paper proposes TimeOmni-VL, a unified vision-centric framework that successfully bridges the long-standing gap between "numerical generation" and "semantic understanding" in the time series field through time-to-image bidirectional mapping (Bi-TSI) and CoT (CoT). During the peer review process, all reviewers unanimously praised the research's exploratory motivation and the newly constructed benchmark suite (TSUMM-Suite). Facing numerous pointed questions from reviewers regarding the practical advantages of the multimodal approach compared to native time series models, the rigorous ablation validation of the CoT semantic mechanism, baseline model comparisons (such as with VisionTS++ on equivalent data), and robustness in complex scenarios such as high dimensionality and variable disorder, the authors provided detailed and convincing supplementary experiments and in-depth analysis during the rebuttal stage. All four reviewers explicitly stated that their concerns had been alleviated and unanimously gave a positive final score (Accept/Weak Accept). Given that this paper represents a significant step forward in the unified modeling of multimodal time series and possesses comprehensive experimental support, it is recommended for acceptance.